# VRPAGENT: LLM-DRIVEN DISCOVERY OF HEURISTIC OPERATORS FOR VEHICLE ROUTING PROBLEMS

## ABSTRACT

Designing high-performing heuristics for vehicle routing problems (VRPs) is a complex task that requires both intuition and deep domain knowledge. Large language model (LLM)-based code generation has recently shown promise across many domains, but it still falls short of producing heuristics that rival those crafted by human experts. In this paper, we propose VRPAGENT, a framework that integrates LLM-generated components into a metaheuristic and refines them through a novel genetic search. By using the LLM to generate problem-specific operators, embedded within a generic metaheuristic framework, VRPAGENT keeps tasks manageable, guarantees correctness, and still enables the discovery of novel and powerful strategies. Across multiple problems, including the capacitated VRP, the VRP with time windows, and the prize-collecting VRP, our method discovers heuristic operators that outperform handcrafted methods and recent learning-based approaches while requiring only a single CPU core. To our knowledge, VRPAGENT is among the first LLM-based paradigms to advance the state-of-the-art in VRPs, highlighting a promising future for automated heuristics discovery.

## 1 INTRODUCTION

Solving combinatorial optimization problems requires sophisticated solution approaches. This is especially true for vehicle routing problems (VRPs), where real-world instances often involve complex constraints and a large number of customers. Over the past decades, operations researchers have developed countless heuristics to address these problems (Konstantakopoulos et al., 2022). Designing a new method that meaningfully improves upon the state of the art across multiple problems is extremely challenging, requiring years of experience and a deep understanding of both general heuristics and the specific problem at hand. Practitioners face similar challenges when applying heuristics. Real-world applications often involve ever-changing requirements that are not supported by existing solution methods. Adapting approaches from the literature to such requirements is a time-consuming and challenging task, and is often considered impractical, even by large companies.

In recent years, neural combinatorial optimization (NCO) has garnered increasing attention due to its potential to automate the discovery of effective heuristics (Bello et al., 2017; Bengio et al., 2021). NCO approaches aim to solve optimization problems by training deep neural networks, typically with reinforcement learning. While NCO methods have demonstrated the ability to learn powerful solution strategies for various combinatorial problems, they also come with notable limitations. First, they require expensive GPUs at test time, which restricts their practical deployment. Second, scalability remains a major challenge as their reliance on attention mechanisms makes it difficult to apply these models to problems that involve processing full distance matrices. Finally, the learned strategies are often difficult for experts to interpret, which raises concerns about their safety and reliability in real-world applications.

The recent advent of performant large language models (LLMs) has enabled new opportunities for automation in general algorithmic design across domains ranging from code synthesis to symbolic planning and mathematical discovery (Madaan et al., 2023). LLMs have proven to be promising approach for discovering new heuristics in combinatorial optimization problems: they can be used to design new heuristics from scratch or adapt existing ones to real-world requirements, enabling customized solutions at a fraction of the cost of an operations research (OR) expert. Among pioneering works, Romera-Paredes et al. (2024); Liu et al. (2024a); Ye et al. (2024a) propose evolutionary

frameworks that iteratively evolve general CO problem heuristics. Recent research has focused on increasingly sophisticated approaches for automating heuristic discovery (Dat et al., 2025; Zheng et al., 2025; Yang et al., 2025b; Novikov et al., 2025; Liu et al., 2025). Although these works provide valuable contributions, the discovered heuristics still fall short of those designed by human experts for VRPs. We identify key limitations in most of these works: design of end-to-end functions, absence of correctness guardrails and overall solution frameworks, and inefficient exploration of the search space, which leads to a failure in challenging the state-of-the-art.

We introduce VRPAGENT, a novel approach that uses LLMs to design heuristic operators for a large neighborhood search (LNS). The high-level LNS is designed to be largely problem-agnostic, allowing our framework to tackle new problems by creating new heuristic operators with minimal human input. To discover strong operators for the LNS, we employ a genetic algorithm (GA) with elitism and biased crossover that iteratively improves operator quality. By focusing on a metaheuristic framework where only problem-specific operators are generated via LLMs, we keep the generation task manageable and effective, while still enabling strong performance on complex problems. We evaluate our method on the capacitated vehicle routing problem (CVRP), the vehicle routing problem with time windows (VRPTW), and the prize-collecting VRP (PCVRP). Our approach discovers heuristic operators for all problems that significantly outperform those designed by human experts.

In summary, we make the following contributions with VRPAGENT:

- We propose an LNS-based metaheuristic in which the problem-specific heuristic operators are generated by an LLM.
- We introduce a simple GA for heuristic discovery featuring elitism with biased crossover for improved exploitation, and a code length penalty to reduce LLM inference costs.
- We show that VRPAGENT discovers strong heuristics across multiple tasks. To the best of our knowledge, it is the first LLM-based approach to advance the state-of-the-art in VRPs.

## 2 RELATED WORK

**Traditional Heuristics** VRPs are ubiquitous problems in logistics that have been studied for decades. Large and richly constrained instances remain difficult to solve to optimality within a practical time frame, and thus heuristics are commonly used in real-world settings (Santini et al., 2023). LNS and its adaptive variants are particularly influential, iteratively destroying and repairing parts of a solution (Shaw, 1998; Schrimpf et al., 2000; Christiaens & Vanden Berghe, 2020). Other well-known approaches include LKH3 (Helsgaun, 2017) and hybrid genetic search (HGS) (Vidal, 2022; Wouda et al., 2024). While capable of producing high-quality solutions, these approaches take significant expertise to design and implement, motivating the need for automating their design.

**Neural Combinatorial Optimization** NCO aims to automate heuristic design by training neural networks from data or via reinforcement learning (Bengio et al., 2021; Berto et al., 2025a; Li et al., 2025b). Approaches can be broadly divided into construction and improvement methods. Construction methods, such as pointer networks (Vinyals et al., 2015; Bello et al., 2017) and subsequent attention-based models for VRPs (Kool et al., 2019; Kwon et al., 2020; Kim et al., 2022; Berto et al., 2025b; Huang et al., 2025a), generate complete solutions quickly in an autoregressive fashion. Further works include enhancements for diversity (Grinsztajn et al., 2023; Hottung et al., 2025a) and out-of-distribution robustness (Drakulic et al., 2023; Luo et al., 2023). Improvement methods instead refine existing solutions at test time, for example, by learning local edits (Ma et al., 2021), guiding $k$-opt moves (Wu et al., 2019; da Costa et al., 2020; Ma et al., 2023), or integrating with metaheuristic approaches as LNS (Hottung & Tierney, 2020) or ant colony optimization (Ye et al., 2023; Kim et al., 2025). Divide-and-conquer frameworks further extend scalability to large instances (Kim et al., 2021; Li et al., 2021; Ye et al., 2024b; Ouyang et al., 2025). Hottung et al. (2025b) adopts a LNS approach with learned heuristics for deconstruction and ordering VRP nodes, showing competitive results against state-of-the-art solvers. Despite continuous progress, most NCO work still falls short of state-of-the-art handcrafted solvers and requires expensive GPU resources, motivating our use of LLM-generated operators as a lightweight alternative.

**Automated Heuristic Discovery** The goal of automatically discovering high-performing heuristics is a long-standing challenge in optimization (Muth, 1963). Early works include genetic pro-

gramming and hyper-heuristics, which construct new solution methods by combining or tuning a set of low-level heuristic components (Burke et al., 2006; 2013) and grammar-based generation (Mascia et al., 2014). The recent advent of LLMs has enabled a new wave of automation for algorithmic design across domains ranging from code synthesis to symbolic planning and mathematical discovery (Madaan et al., 2023; Shinn et al., 2023; Novikov et al., 2025). Early works in heuristic discovery with LLMs including Romera-Paredes et al. (2024); Liu et al. (2024a) employ evolutionary approaches that generate heuristic code snippets for simple heuristics in combinatorial problems, including VRPs, packing, and scheduling. Building on this trend, reflection-augmented evolution has been shown to discover more sophisticated heuristics during the refinement process (Ye et al., 2024a). Orthogonal search strategies further expand the design space: diversity-driven evolution (Dat et al., 2025), Monte Carlo tree search (Zheng et al., 2025), ensembling of different LLMs (Novikov et al., 2025), meta-prompt optimization (Shi et al., 2025), and portfolio-style discovery of sets of complementary heuristics (Yang et al., 2025b; Liu et al., 2025). More specifically for VRPs, Tran et al. (2025) design heuristics to enhance NCO model decoding. In parallel, finetuning and instruction specialization of LLMs for algorithmic synthesis have been proposed to improve reliability and sample efficiency (Šurina et al., 2025; Huang et al., 2025b; Chen et al., 2025b), and benchmark suites have begun to standardize evaluation protocols for LLM-driven heuristics (Liu et al., 2024b; Sun et al., 2025; Feng et al., 2025; Li et al., 2025a; Chen et al., 2025a).

Despite encouraging progress, LLM-generated heuristics still lag behind traditional solvers and NCO methods alike on VRPs, especially under tight time budgets and realistic constraints. We identify three recurring limitations: (i) weak "agentic playground" formulations that ask LLMs to design small heuristic snippets without an overall solution framework; (ii) weak or absent correctness guards around generated code; and (iii) inefficient exploration that drifts toward verbose, brittle implementations. Our approach follows the principle of *keeping AI agents on a leash*: we constrain the search to problem-specific operators nested within a robust, correctness-enforcing metaheuristic. VRPAGENT's design keeps the synthesis task tractable, preserves feasibility, and still enables the discovery of novel operators that can advance the state-of-the-art for VRP solving.

## 3 VEHICLE ROUTING PROBLEMS

VRPs are a fundamental class of combinatorial optimization problems with the aim to minimize travel costs while respecting some constraints. Travel cost is usually measured by the total distance traveled. Formally, a VRP is defined on a graph $G = (V, E)$, where each node $i \in V$ denotes a customer and each edge $(i, j) \in E$ models traveling from $i$ to $j$ with an associated cost, e.g., the distance between $i$ and $j$. All routes originate from and end at the depot node $0$. In the **CVRP**, vehicles performing the routes have a limited capacity. The total demand on any route cannot exceed the vehicle's capacity $C$ at any time, and every customer is served exactly once. The **VRPTW** extends this setting, assigning a service time $s_i$ and a time window $[t_i^l, t_i^r]$ to each customer $i$. The service to any customer must start within their time window, i.e., if a vehicle arrives before a customer's time window starts, it has to wait until $t_i^l$. The **PCVRP** relaxes the requirement of visiting all customers. Servicing a customer $i$ is associated with a prize $p_i$. The objective is to maximize the total collected prize while minimizing travel cost.

## 4 VRPAGENT

VRPAGENT is a framework for solving VRPs that automatically discovers strong heuristic operators using LLMs. It is built on two main components. The first (Section 4.1) is an LNS (Shaw, 1998) variant for VRPs, which relies on heuristic operators to improve solutions iteratively. At test time, this LNS produces solutions for VRP instances on a single CPU core. The second component (Section 4.2) is a GA used in a discovery phase to generate these heuristic operators. In the discovery phase, operator implementations are iteratively created, modified, and refined with the help of an LLM. Classic genetic operations such as crossover and mutation are applied to operator implementations, with the LLM carrying out these transformations. Each generated operator is evaluated by inserting it into the LNS and testing the resulting search performance on a set of training instances. The performance on these instances defines the fitness value of the individual. Over successive generations, the GA produces increasingly effective heuristic operators. Fig. 1 shows a high-level overview of VRPAGENT.

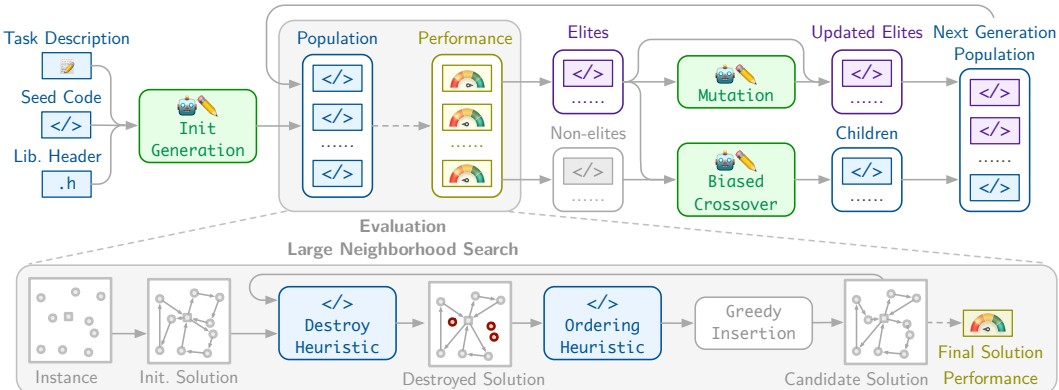

Figure 1: VRPAGENT overview.

## 4.1 LARGE NEIGHBORHOOD SEARCH WITH LLM-GENERATED OPERATORS

VRPAGENT employs an LNS with LLM-generated operators to find solutions for VRPs. The high-level LNS guides the search process and ensures feasibility by acting as a safeguard around the LLM-generated code. In short, the LNS works by repeatedly removing a set of customers from their tours, ordering the removed customers, and reinserting them one by one at their locally optimal positions. The removal and ordering strategies are defined by LLM-written heuristic operators.

Algorithm 1 presents the pseudocode of our LNS. The algorithm begins by generating an initial solution $s$ for a given instance $l$. For all routing problems, this initial solution is constructed with one tour per customer. The solution is then iteratively improved until a termination criterion is met. In each iteration, $s$ is first destroyed by removing customers from their tours using the LLM-generated removal operator $f_{\text{REMOVE}}$. This yields an incomplete solution $s'$ in which the removed customers are unassigned. Next, the removed customers are ordered by the LLM-generated operator $f_{\text{ORDER}}$. They are then reinserted one by one in that order, always placed at the locally best position. That is, the insertion that increases the objective value as little as possible. Finally, an acceptance decision is made: $s'$ may replace $s$ only if it is better, or it may be accepted under a simulated annealing rule. After all iterations, the algorithm returns the best solution $s$.

---

**Algorithm 1** VRPAGENT-LNS

**Input:** CVRP Instance $l$, Destroy Operator $f_{\text{REMOVE}}$, Ordering Operator $f_{\text{ORDER}}$

1: **function** LNS($l$, $f_{\text{REMOVE}}$, $f_{\text{ORDER}}$)
2:    $s \leftarrow$ GENERATESTARTSOLUTION($l$)
3:    **while** termination criteria not reached **do**
4:        $s' \leftarrow f_{\text{REMOVE}}(l, s)$                    ▷ Remove some customers from their tours (**LLM-Operator**)
5:        $insertionOrder \leftarrow f_{\text{ORDER}}(l, s')$              ▷ Order the removed customers (**LLM-Operator**)
6:        **for** $c$ in $insertionOrder$ **do**                  ▷ Reinsert removed customers one by one
7:            Insert customer $c$ at their locally optimal position in $s'$
8:        **end for**
9:        $s \leftarrow$ ACCEPT($s, s'$)
10:    **end while**
11:    **return** $s$
12: **end function**

---

## 4.2 HEURISTIC DISCOVERY

VRPAGENT discovers heuristic operators using a simple GA that is strongly geared toward exploitation. Given the very large search space and limited search budget, this bias toward exploitation proves highly beneficial, leading to significant improvements in our experiment. Each individual in our GA represents an implementation of the operator pair $(f_{\text{REMOVE}}, f_{\text{ORDER}})$ as C++ code. During

the discovery phase, new individuals are created through the means of mutation and crossover. To evaluate an individual, we run VRPAGENT-LNS using its operator pair on a set of training instances.

Algorithm 2 outlines the core logic of our GA. It takes as input the initial population size $M_{\text{init}}$, the number of elites $M_{\text{E}}$, and the number of offspring $M_{\text{C}}$ generated in each iteration. The algorithm begins by creating an initial population (i.e., a set) of heuristic operators and then enters the main evolutionary loop, which runs until a termination criterion is reached. At the start of each iteration, all individuals are evaluated, and the top $M_{\text{E}}$ are placed in the set of elites $\mathcal{P}_E$, and the reminder in the set of non-elites $\mathcal{P}_{NE}$. Next, $M_{\text{C}}$ offspring are created by pairing one elite with one non-elite individual and combining them using biased crossover. This crossover favors the elite parent while still injecting diversity from the non-elite.

---

**Algorithm 2** VRPAGENT-GA

**Input:** Initial population size $M_{\text{init}}$, number of elites $M_{\text{E}}$, number of offspring $M_{\text{C}}$

1: **function** GA($M_{\text{init}}, M_{\text{E}}, M_{\text{C}}$)
2:     $\mathcal{P} \leftarrow$ GENERATESTARTPOP($M_{\text{init}}$)                          ▷ Initialize population
3:     **while** termination criteria not reached **do**
4:         $(\mathcal{P}_E, \mathcal{P}_{NE}) \leftarrow$ TOP-K-ELITE($\mathcal{P}, M_{\text{E}}$)     ▷ Rank heuristics and take the top $M_{\text{E}}$ as elite
5:         $C \leftarrow \emptyset$
6:         **while** $|C| < M_{\text{C}}$ **do**
7:             $p_e \leftarrow$ RANDOM($\mathcal{P}_E$)                          ▷ Select random elite
8:             $p_{ne} \leftarrow$ RANDOM($\mathcal{P}_{NE}$)                     ▷ Select random non-elite
9:             $p_c \leftarrow$ BIASED-CROSSOVER($p_e, p_{ne}$)              ▷ Generate new heuristic via crossover
10:            $C \leftarrow C \cup \{p_c\}$
11:        **end while**
12:        **for all** $p_e \in \mathcal{P}_E$ **do**                          ▷ Improve each elite via mutation
13:            $p_m \leftarrow$ MUTATION($p_e$)                          ▷ Modify heuristic using mutation prompt
14:            **if** FIT($p_m$) < FIT($p_e$) **then**                     ▷ Replace elite if improved
15:                $p_e \leftarrow p_m$
16:            **end if**
17:        **end for**
18:        $\mathcal{P} \leftarrow \mathcal{P}_E \cup C$                          ▷ Create next generation
19:    **end while**
20:    **return** BEST($\mathcal{P}$)
21: **end function**

---

Each elite is refined through mutation. Unlike standard GAs, where mutation is typically applied to offspring to promote exploration, we apply it directly to elites. These mutations are small, making the procedure more exploitation-focused. If a mutated elite achieves better fitness, it replaces the original; otherwise, the original is kept. This replacement rule ensures that mutated elites do not accumulate in the population, thereby preventing premature convergence. Finally, the next generation is formed by combining the elites with the newly generated offspring. The process repeats until termination, after which the best heuristic operator discovered is returned. We describe the initialization, crossover, mutation, and fitness evaluation steps in the following paragraphs.

**Initial Population Generation**    The initial population is created by prompting the LLM to generate implementations of the two operators. For this, the LLM is provided with a global system context that explains the overall problem and the LNS, a trivial example implementation of both operators, and technical details of the LNS implementation in the form of C++ header files that allow operators to access shared variables and methods efficiently (see Prompts 1, 2 and 9 in Appendix A).

**Biased Crossover**    The offspring are produced by combining two parents through crossover. We use biased crossover, which pairs an elite individual with a non-elite one. The LLM is given both their implementations and is prompted to take most ideas and concepts from the implementation of the elite individuals and only a predefined % from the implementation of the non-elite individual. This adaptation of standard crossover significantly increases exploitation. The complete crossover instruction provided to the LLM corresponds to Prompts 1 and 3 in Appendix A.

**Mutation**    VRPAGENT uses mutation to slightly modify elite implementations. Following Liu et al. (2024a), we implement multiple mutation prompts that focus on different areas of improvement

for better exploration. More precisely, the LLM is given the implementation that should be modified together with one randomly selected mutation prompt (Prompts 4 to 7 in Appendix A) and the global system context Prompt 1. The four supported mutations include *Ablation* (i.e., removing a random mechanic), *Extend* (i.e., adding a new mechanic), *Adjust-Parameters* (i.e., changing the hyperparameter settings), and *Refactor* (i.e, modify the code so that the runtime is improved).

**Fitness Function with Code Length Penalty**   We evaluate an individual $i$ by running VRPA-GENT-LNS with the operator pair defined by $i$ on a set of training instances $I^{\text{train}}$. The resulting solutions are then used to compute the fitness value. Specifically, the fitness of $i$ is defined as the average objective value across all training instances, plus a penalty proportional to the length (i.e., number of lines) of the corresponding implementation $\mathcal{C}_i$:

$$\text{Fit}(i) \;=\; \frac{1}{|I^{\text{train}}|} \sum_{j \in I^{\text{train}}} \text{Obj}(s_{i,j}) \;+\; \lambda \cdot \text{Len}(\mathcal{C}_i), \tag{1}$$

where $s_{i,j}$ is the solution obtained by applying VRPAGENT-LNS with the operators of individual $i$ to training instance $j$, and $\lambda$ controls the strength of the code length penalty.

The penalty helps prevent uncontrolled growth in implementation size, which we observed when no regularization was applied. More compact implementations are also easier for humans to interpret and maintain, making the generated heuristics more useful in practice. Finally, a shorter code reduces the number of tokens processed by the LLM during generation, which in our experiments lowers token usage by more than 50%, and thus significantly reduces generation costs and latency.

## 5   EXPERIMENTS

We evaluate VRPAGENT on three vehicle routing problems: the CVRP, the VRPTW, and the PCVRP. The best discovered heuristics are made publicly available in our online repository at https://anonymous.4open.science/r/vrpagent-submission.

**VRPAGENT Hyperparameters**   During the discovery phase we use the following parameters unless stated otherwise: elite size $M_E = 10$, offspring size $M_C = 30$, an initial population size of $M_{\text{init}} = 100$. The code length penalty factor is set to $\lambda = 2 \cdot 10^{-4}$ and the discovery phase is terminated after 40 iterations. Individuals are evaluated with VRPAGENT-LNS on a training set of 64 instances, each with 500 customers, using a runtime limit of 20s per instance. We employ Gemini 2.5 Flash (Comanici et al., 2025) as the LLM.

### 5.1   COMPARISON TO STATE-OF-THE-ART

**Benchmark Setup**   We evaluate all methods on the three problems using instance sets of 500, 1000, and 2000 customers. To ensure consistency, we adopt the same test instances and baseline configurations as described in Hottung et al. (2025b) using a single core of an AMD Milan 7763 processor and an additional single NVIDIA A100 for approaches that require a GPU. For a fair comparison, we limit the search by runtime when possible. The operators used by VRPAGENT are obtained from 10 discovery runs per problem (conducted on instances of size 500 only), with the best operator selected based on performance on a separate validation set. Additionally, we provide an evaluation of VRPAGENT on the more realistic X instances in Appendix C, which better reflect the properties encountered in real-world scenarios.

**Baselines**   We compare VRPAGENT to several established operations research solvers: HGS (Vidal, 2022), SISRs (Christiaens & Vanden Berghe, 2020), LKH3 (Helsgaun, 2017), and Gurobi (Gurobi Optimization, LLC, 2024). We also include PyVRP (Wouda et al., 2024) (version 0.9.0), an open-source extension of HGS that supports additional VRP variants, and the recent GPU-based NVIDIA cuOpt (NVIDIA Corporation, 2025). For the CVRP, we further consider learning-based approaches that require GPUs at test time: BQ (Drakulic et al., 2023), LEHD (Luo et al., 2023), UDC (Zheng et al., 2024b), and NDS (Hottung et al., 2025b). In addition, we compare against LLM-based methods that learn a construction heuristic, including EoH (Liu et al., 2024a), MCTS-AHD (Zheng et al., 2024a), and ReEvo (Ye et al., 2024a). These approaches only generate a single

solution with runtime values <1 second and do not benefit from additional search budget. We further compare against three stronger LLM-based baselines that leverage search: ReEvo-ACO, which combines LLM-generated heuristics with the Ant Colony Optimization (ACO) metaheuristic, NCO-LLM (Tran et al., 2025), which enhances LEHD by automating the design of logit reshaping during search, and LLM-LNS (Ye et al., 2025), a recent method using LLMs to design search heuristics.

**Results** Table 1 presents the results of our experiments. On instances with 1000 and 2000 customers, VRPAGENT outperforms all other methods across all problem types, achieving gaps of around $-0.30\%$ relative to the state-of-the-art SISRs. This represents a substantial improvement that can translate into significant savings in large-scale, real-world scenarios.

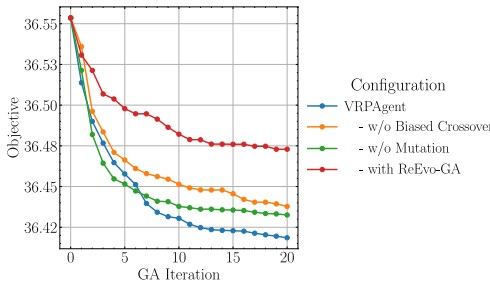

Figure 2: Ablation results.

On smaller instances, VRPAGENT approaches the performance of NDS, which relies on an expensive GPU at test time and is trained specifically for each instance size. Compared to other LLM-based methods, VRPAGENT consistently demonstrates significantly better performance across all test cases. It is also noteworthy that VRPAGENT performs well on all problem sizes, despite being trained only on instances with 500 customers. This strong generalization performance stems from the algorithmic operators. These usually do not rely on scale-dependent neural representations, which are poisoned to implicitly encode all characteristics of the training distribution, including unwanted characteristics like node count and spatial density.

## 5.2 ANALYSES

**Ablation Studies** We analyze the contribution of key components in our GA by disabling or replacing them. Specifically, we test three variants: (i) replacing our biased crossover with a standard crossover, where the LLM is instructed to take roughly half the elements from each parent, (ii) removing mutation while increasing offspring size to maintain a comparable population, and (iii) replacing our entire GA with the GA of ReEvo (Ye et al., 2024a), which uses a reflection mechanism. Each variant is tested on the CVRP 10 times, and results are averaged. Fig. 2 reports performance on the training set during the discovery phase. Across all cases, modifications lead to reduced performance. Biased crossover is particularly important: by favoring elite solutions while still incorporating elements from weaker parents, it balances exploitation and exploration and drives faster convergence. Removing mutation lowers final solution quality, and replacing our GA with ReEvo's yields the weakest results, confirming that our combination of elitism, biased crossover, and mutation is essential for discovering high-quality heuristics.

**Performance Across Different LLMs** We study the performance of VRPAGENT when paired with different LLMs. We conduct discovery runs of 20 iterations each for the CVRP using six models. We access Gemini 2.0 Flash and Gemini 2.5 Flash (Comanici et al., 2025) via API, while Qwen3 (Yang et al., 2025a), Llama 3.3 (Grattafiori et al., 2024), Gemma 3 (Team et al., 2025), and gpt-oss (Agarwal et al., 2025) are served locally via vLLM (Kwon et al., 2023). Fig. 3 reports the average objective value on the training set during discovery (left) and the total computational cost per run (right). All tested models substantially improve the heuristic operators throughout the discovery process. Gemini 2.5 Flash and gpt-oss both discover heuristics that outperform the state-of-the-art baseline. Gemini 2.5 Flash achieves the best overall results, but at a cost of nearly $20 per run. In contrast, the open-source gpt-oss model, run on two NVIDIA A100 (40GB) GPUs, achieves nearly the same performance under $2 per run.

**Performance Over the Discovery Process** We analyze the convergence rate of the discovery process with Gemini 2.5 Flash on all three problems across 40 iterations. As a baseline, we report the performance of VRPAGENT-LNS when used in combination with handcrafted operators. Specifically, we reimplement the operators from SISRs (Christiaens & Vanden Berghe, 2020), which represent the state of the art in LNS-based routing methods. As shown in Fig. 4, VRPAGENT produces heuristics that outperform the state-of-the-art (SOTA) handcrafted operators. In Appendix D, we ad-

Table 1: Performance on test data. The gap is calculated relative to SISRs. Runtime is reported on a per-instance basis in seconds. The best results (i.e., those with the lowest objective function value) are shown in **bold**, and the second-best are underlined. * Indicates that a feasible solution was not found for all instances. OOM indicates that the corresponding solver yielded an out-of-memory error. VRPAGENT demonstrates state-of-the-art results among heuristics and NCO solvers at larger scales.

| | Method | | N=500 | | | N=1000 | | | N=2000 | | |
|---|---|---|---|---|---|---|---|---|---|---|---|
| | | | Obj.↓ | Gap↓ | Time | Obj.↓ | Gap↓ | Time | Obj.↓ | Gap↓ | Time |
| **CVRP** | SISRs | CPU | 36.65 | - | 60 | 41.14 | - | 120 | 56.04 | - | 240 |
| | HGS | CPU | 36.66 | 0.00% | 60 | 41.51 | 0.84% | 121 | 57.38 | 2.33% | 241 |
| | LKH3 | CPU | 37.25 | 1.66% | 174 | 42.16 | 2.46% | 408 | 58.12 | 3.70% | 1448 |
| | NVIDIA cuOpt | CPU+GPU | 37.38 | 1.98% | 60 | 42.71 | 3.78% | 121 | 59.22 | 5.66% | 241 |
| | Gurobi | CPU | 47.85 | 30.53% | 60 | 87.36 | 111.88% | 120 | - | OOM | - |
| | BQ (BS64) | CPU+GPU | 37.51 | 2.34% | 23 | 43.32 | 5.30% | 164 | - | - | - |
| | LEHD (RRC) | CPU+GPU | 37.04 | 1.06% | 60 | 42.47 | 3.25% | 121 | 60.11 | 7.25% | 246 |
| | UDC | CPU+GPU | 37.63 | 2.69% | 60 | 42.65 | 3.68% | 121 | - | - | - |
| | NDS | CPU+GPU | 36.57 | **-0.20%** | 60 | 41.11 | -0.07% | 120 | 56.00 | -0.07% | 240 |
| | EoH | CPU | 45.89 | 25.21% | <1 | 52.42 | 27.42% | <1 | 71.21 | 27.07% | <1 |
| | MCTS-AHD | CPU | 45.51 | 24.17% | <1 | 52.49 | 27.59% | <1 | 71.15 | 26.96% | <1 |
| | ReEvo | CPU | 44.21 | 20.63% | <1 | 52.23 | 26.96% | <1 | 70.01 | 24.93% | <1 |
| | ReEvo-ACO | CPU | 40.25 | 9.83% | 60 | 46.22 | 12.34% | 120 | 63.76 | 13.77% | 240 |
| | NCO-LLM | CPU+GPU | 36.93 | 0.76% | 60 | 41.96 | 1.99% | 121 | 59.43 | 6.05% | 246 |
| | LLM-LNS | CPU | 38.99 | 6.38% | 60 | 45.00 | 9.38% | 120 | 62.28 | 11.13% | 240 |
| | **VRPAGENT** | CPU | 36.60 | -0.12% | 60 | 41.06 | **-0.19%** | 120 | 55.98 | **-0.11%** | 240 |
| **VRPTW** | SISRs | CPU | 48.09 | - | 60 | 87.68 | - | 120 | 167.49 | - | 240 |
| | PyVRP-HGS | CPU | 49.01 | 1.91% | 60 | 90.35 | 3.08% | 120 | 173.46 | 3.62% | 240 |
| | NVIDIA cuOpt | CPU+GPU | 49.30 | 2.60% | 61 | 90.31 | 3.11% | 121 | 173.52 | 3.85%* | 243 |
| | Gurobi | CPU | 69.01 | 43.83% | 60 | 148.45 | 71.46% | 120 | - | OOM | - |
| | NDS | CPU+GPU | 47.94 | **-0.30%** | 60 | 87.54 | -0.16% | 120 | 167.48 | -0.00% | 240 |
| | EoH | CPU | 60.40 | 25.60% | <1 | 118.80 | 35.49% | <1 | 245.70 | 46.70% | <1 |
| | MCTS-AHD | CPU | 58.31 | 21.25% | <1 | 113.72 | 29.70% | <1 | 231.11 | 37.98% | <1 |
| | ReEvo | CPU | 58.01 | 20.63% | <1 | 110.55 | 26.08% | <1 | 218.90 | 30.69% | <1 |
| | ReEvo-ACO | CPU | 52.91 | 10.03% | 60 | 97.39 | 11.07% | 120 | 193.13 | 15.31% | 240 |
| | **VRPAGENT** | CPU | 47.97 | -0.24% | 60 | 87.40 | **-0.33%** | 120 | 166.96 | **-0.33%** | 240 |
| **PCVRP** | SISRs | CPU | 43.22 | - | 60 | 81.12 | - | 120 | 158.17 | - | 240 |
| | PyVRP-HGS | CPU | 44.97 | 4.10% | 60 | 84.91 | 4.81% | 120 | 165.56 | 4.78% | 240 |
| | NVIDIA cuOpt | CPU+GPU | 43.34 | 0.19% | 60 | 81.89 | 0.84% | 121 | 160.33 | 1.22% | 241 |
| | Gurobi | CPU | 71.58 | 68.23% | 60 | 147.08 | 86.03% | 120 | - | OOM | - |
| | NDS | CPU+GPU | 43.12 | **-0.23%** | 60 | 80.99 | -0.17% | 121 | 158.09 | -0.06% | 241 |
| | **VRPAGENT** | CPU | 43.18 | -0.09% | 60 | 80.95 | **-0.21%** | 120 | 157.69 | **-0.32%** | 240 |

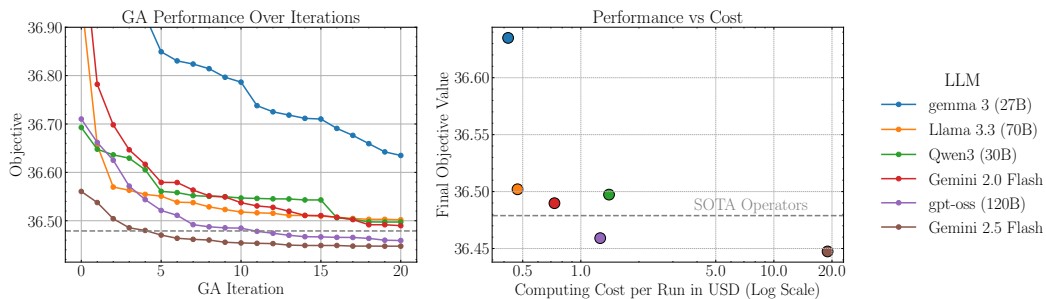

Figure 3: Performance on the CVRP for different LLMs. Detailed cost calculations are provided in Section B.1.

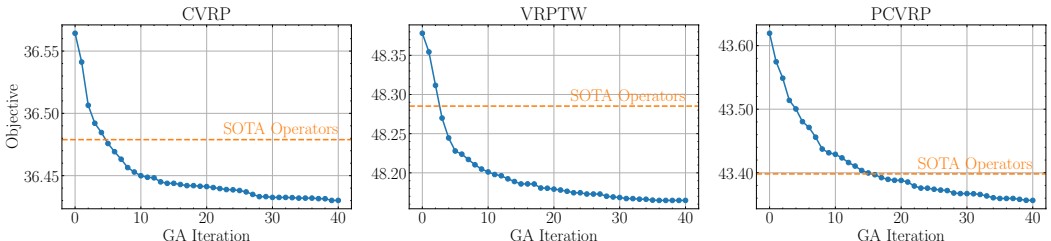

Figure 4: Performance over the course of the discovery process.

ditionally report the percentage of operators that compile successfully and produce valid solutions (success rate) throughout the runs.

**Crossover Bias and Elite Size** We evaluate the effect of the crossover bias and elite size $M_E$ on the performance of our GA. As shown in Fig. 5a, elite sizes between 5 and 15 yield strong performance, whereas an elite size of 1 leads to drastically worse results by making the search overly greedy. Regarding the crossover bias, a strong preference for the better-performing operator is clearly beneficial, with an operator bias of 80% achieving the best results.

**Code Length Penalty** We investigate the impact of the code length penalty factor $\lambda$ on both the quality and length of the discovered heuristics. Several discovery runs with varying $\lambda$ values reveal that the penalty strongly controls implementation size without substantially degrading performance. For instance, increasing $\lambda$ to $4 \cdot 10^{-4}$ reduces the average length of generated heuristics by roughly 50%, while only causing a marginal drop in objective value (Fig. 5b). These results highlight several key insights. First, the penalty effectively discourages overly long implementations, producing heuristics that are easier for humans to read and maintain. Second, higher penalties reduce token usage during LLM generation, improving efficiency and lowering computational costs. Third, the fact that performance remains largely unaffected even for strong penalties demonstrates that simple, concise heuristics can perform just as well as more complex ones.

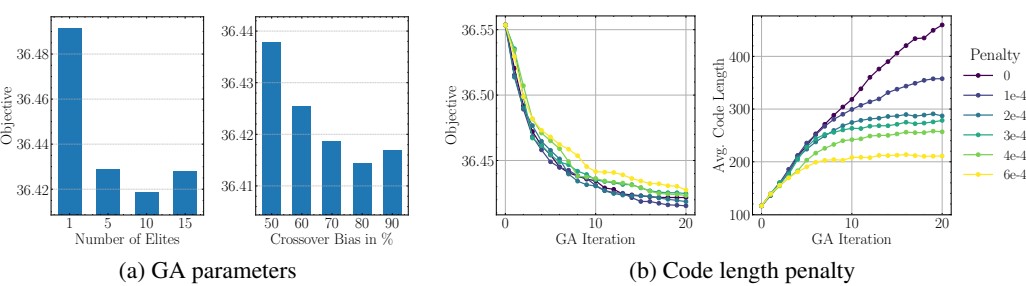

(a) GA parameters             (b) Code length penalty

Figure 5: Results of the sensitivity analyses.

**Warm-starting Heuristic Discovery**    We analyze whether heuristics discovered for one problem can be adapted to others via warm-starts. Specifically, we initialize the discovery process for the VRPTW and PCVRP using the best CVRP heuristic and task the model with adapting it to the new problem. Warm-starting consistently accelerates convergence and improves final performance compared to cold-start runs, making this approach promising for real-world settings where related problems need to be solved efficiently. Detailed results and analyses are provided in Appendix E.

## 6    DISCOVERED HEURISTIC OPERATORS

A key advantage of LLM-driven heuristic discovery over deep learning–based approaches is that the resulting strategies can be directly inspected and interpreted by human experts. To further facilitate such analysis, we introduce a post-processing stage in which an LLM is used to improve the code quality and readability of the generated implementations. This stage is necessary because code quality is not considered during the discovery process itself: the fitness functions focus solely on performance, and we explicitly instruct the LLMs not to include comments during search. While this reduces interpretability, it substantially lowers token usage and accelerates the discovery procedure.

**Code quality improvement**    To refine the quality of the discovered heuristics, we pass each implementation to an LLM together with a prompt specifying the desired improvements. The returned code is then evaluated on the validation instances to verify that performance remains unchanged. We repeat this refinement–evaluation loop until we obtain an improved version that matches the original performance. The improvement prompt emphasizes (1) adding comments that explain both high-level heuristic logic and low-level implementation details, (2) restructuring code into clearer functional units, (3) and eliminating unnecessarily convoluted uses of random numbers.

**Analysis**    We give the task of analyzing the best discovered heuristic operators for each problem to three coauthors of this paper who have years of experience writing OR heuristics in routing and related fields. The goal of our analysis is to assess the (1) readability, (2) coherence and soundness, (3) maintainability, (4) interpretability, (5) and novelty of the generated heuristic operators. We note that our assessment of the heuristics is not meant to be a thorough scientific analysis that generalizes to other LLMs or optimization problems. We offer a detailed analysis in Appendix F.

The removal and sorting mechanisms of all three analyzed heuristics can be described as ensemble approaches that use random numbers to choose different (combinations of) heuristics in each iteration. Given the popularity and success of such ensembles in well-known metaheuristics (e.g., adaptive LNS (Pisinger & Ropke, 2018)), it is perhaps not a surprise that we encounter ensembles in the discovered heuristics. Overall, we find the generated heuristics to be relatively easy to interpret and structurally coherent. The comments introduced during post-processing substantially enhance comprehensibility, and the decomposition of the code into functions further improves readability.

All of the heuristics generated can be said to be novel. We are not aware of any heuristics in the literature exactly matching these algorithms, however we note that the heuristics mainly consist of recombinations of ideas existing in the literature, e.g., SISRs or simple greedy criteria related to distance/demand/time/prizes. Given the complexity of the ensembles, with some having up to nine different component heuristics, a detailed ablation study would be necessary to try to find out which components or combinations of components lead to good performance.

## 7    CONCLUSION

In this work, we introduced VRPAGENT, a metaheuristic framework in which LLMs generate problem-specific operators for a LNS. By focusing on operator generation rather than end-to-end heuristics, VRPAGENT makes the discovery task more manageable while achieving strong performance. Using a GA with elitism and biased crossover for algorithm discovery, VRPAGENT consistently finds heuristic operators that outperform human-designed approaches on a range of vehicle routing problems. Our results highlight a promising future for automated heuristic discovery, suggesting that LLMs could play a key role in designing efficient and adaptable optimization methods for complex, real-world problems. For future work, we will investigate how to further simplify the generated heuristics to help increase the ease of using VRPAGENT generated code in practice.

REPRODUCIBILITY STATEMENT

We have made every effort to ensure the reproducibility of our results. Detailed descriptions of configurations, prompts, the discovery pipeline, and overall experimental setups are provided in both the main paper and the appendix to enable independent reproducibility. All code to reproduce the experiments will be made open-source upon acceptance.

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

# A  PROMPTS

The exact prompts used by VRPAGENT-GA are presented below. We distinguish between *general prompts*, which remain the same across all problems, and *problem-specific prompts*, which must be tailored to each task. The two are combined by substituting variables in the general prompts (e.g., replacing {problem_desc} with the corresponding problem-specific description). For crossover and ablation prompts, the provided template is further extended by inserting the implementations of the associated individuals at the designated positions.

For brevity, we report only the problem-specific prompts for the CVRP here. The prompts for all problems can be found in our online repository.

## A.1  GENERAL PROMPTS

```
You are an operations research expert. Your task is to design new heuristics for an
existing **Large Neighborhood Search (LNS)** framework applied to the
{problem_name_long}. The framework iteratively improves a given initial solution
through the following steps:
1. **Customer Removal**: Select a subset of customers to remove using a specified
heuristic.
2. **Solution Perturbation**: Remove the selected customers from their tours. This
results in an infeasible solution where the removed customers are no longer served.
3. **Customer Ordering**: Order the removed customers using another heuristic.
4. **Greedy Reinsertion**: Reinsert the removed customers one by one into the tours,
following the order defined in step 3.

Your job is to implement **new heuristics for:**
- **Step 1**: Customer selection (`select_by_llm_1`)
- **Step 3**: Ordering of the removed customers (`sort_by_llm_1`)

All other components of the LNS framework are fixed and **cannot be modified**.

# Routing Problem Description
{problem_desc}

# Other implementation notes and requirements:
- The framework is implemented in **C++**.
- The LNS targets **large instances** (e.g., more than 500 customers).
- Only a small number of customers should be removed in each iteration.
- The selected customers do **not need to form a single compact cluster**, but **each
selected customer should be close to at least one or a few other selected customers**.
This encourages meaningful changes during greedy reinsertion.
- The heuristic must incorporate **stochastic behavior** to ensure sufficient diversity
over **millions of iterations**.
- The search is limited by runtime, meaning that the two new heuristics should be very
fast.

# Code style
- IMPORTANT: DO NOT ADD ***ANY*** COMMENTS unless asked
```

Prompt 1: Global system context.

```
[TASK]
Write high-quality heuristics for `select_by_llm_1` and `sort_by_llm_1` in the LNS
framework. Write the full code file in a ```cpp``` code block.

# Example implementation
{seed_code}

# Libary context
You are also provided with some selected header function information with comments that
could be useful:
{LNS_headers}
```

Prompt 2: Initial operator generation.

```
[Better Code]
{code_parent_1}

[Worse Code]
{code_parent_2}

[Task]
Write new high-quality heuristics for `select_by_llm_1` and `sort_by_llm_1` in the LNS
framework. Your implementation
should be a crossover of the two implementations above, taking most ideas from the
better code (80%) and only some ideas from the worse code (20%).
Ensure that the new code maintains a comparable overall complexity and length to the
two implementations above.
Output code only and enclose your code with C++ code block: ```cpp ... ```. Do not
comment your code.
```

Prompt 3: Crossover prompt with 80% bias.

```
[Code]
{code}

[Task]
To simplify the heuristics implemented in  `select_by_llm_1` and `sort_by_llm_1` we want
to conduct an ablation study.
Choose a random mechanic/component from the code that you think might not be important
and remove any trace of it from the code. We will
then run your code to evaluate the impact of the removed component. Output code only
and enclose your code with C++ code block: ```cpp ... ```.
```

Prompt 4: Ablation (Mutation).

```
[Code]
{code}

[Task]
The goal is improve the heuristics implemented in `select_by_llm_1` and `sort_by_llm_1`.
Add a new mechanic/component to the code above. Be innovative. We will
then run your code to evaluate the impact of the new component. Output code only and
enclose your code with C++ code block: ```cpp ... ```.
```

Prompt 5: Extend (Mutation).

```
[Code]
{code}

[Task]
The goal is to find new parameter settings for heuristics implemented in
`select_by_llm_1` and `sort_by_llm_1`.
Modify the parameters of the code above to improve the effectiveness of the heuristic.
If there are magic numbers in the code, replace them with constants that are set at the
beginning of each function.
Do not make any other changes to the code.
Output code only and enclose your code with C++ code block: ```cpp ... ```.
```

Prompt 6: Adjust-Parameters (Mutation).

```
[Code]
{code}

[Task]
The goal is improve the runtime of the heuristics implemented in `select_by_llm_1` and
`sort_by_llm_1`.
Modify the code so that the runtime is reduced. It is ok to slightly change the logic
of the heuristic to achieve this.
Output code only and enclose your code with C++ code block: ```cpp ... ```.
```

Prompt 7: Refactor (Mutation).

## A.2 PROBLEM-SPECIFIC PROMPTS

### A.2.1 CVRP

```
The Capacitated Vehicle Routing Problem (CVRP) involves determining a set of delivery
routes from a depot to a group of customers, where each customer has a specific demand
and each vehicle has a fixed capacity. The objective is to design routes that minimize
the total distance traveled, while ensuring that:
Each route starts and ends at the depot.
Each customer is visited exactly once by a single vehicle.
The total demand on any route does not exceed the vehicle capacity.

There is no limit on the number of vehicles that can be used.
```

Prompt 8: Problem description CVRP).

```
From `Instance.h`:

```cpp
struct Instance {
    int numNodes; // Total number of nodes including depot
    int numCustomers; // Total number of customers (excluding depot)
    int vehicleCapacity; // Capacity of the vehicle (identical for all vehicles)
    std::vector<int> demand;  // Demand of each node (with the depot at index 0 having a
     demand of 0)
    std::vector<std::vector<float>> distanceMatrix; //Distance matrix between nodes
    std::vector<std::vector<float>> nodePositions; // Node positions in 2D space
    std::vector<std::vector<int>> adj; // Adjacency list for each node, sorted by
     distance
}
```

From `Solution.h`:

```cpp
struct Solution {
    const Instance& instance; // Reference to the instance to avoid copying
    float totalCosts; // Total cost of the solution
    std::vector<Tour> tours; // List of tours in the solution
    std::vector<int> customerToTourMap; // Map from each customer to its tour index. This
     can be used to
    // quickly find which tour a customer belongs to, e.g. solution.tours[solution.
     customerToTourMap[c]] returns the tour of customer c.
}
```

From `Tour.h`:

```cpp
struct Tour {
    std::vector<int> customers; // Customers in the tour, excluding depot
    int demand = 0; // Total demand of the tour
    float costs = 0;  // Total cost of the tour including distance to and from the depot
}
```

From `Utils.h`:
```cpp
int getRandomNumber(int min, int max);
float getRandomFraction(float min = 0.0, float max = 1.0);
float getRandomFractionFast(); // Function to generate a random fraction (float) in the
    range [0, 1] using a fast method
std::vector<int> argsort(const std::vector<float>& values); // Function to perform
    argsort on a vector of float values
```
```

Prompt 9: Metaheuristic context ({LNS_headers}).

```cpp
#include "AgentDesigned.h"
#include <random>
#include <unordered_set>
#include "Utils.h"

// Customer selection
std::vector<int> select_by_llm_1(const Solution& sol) {
    // random selection of customers
        std::unordered_set<int> selectedCustomers;

        int numCustomersToRemove = getRandomNumber(10, 20);

        while (selectedCustomers.size() < numCustomersToRemove) {
            int randomCustomer = getRandomNumber(1, sol.instance.numCustomers);
            selectedCustomers.insert(randomCustomer);
        }

        return std::vector<int>(selectedCustomers.begin(), selectedCustomers.end());
}

// Function selecting the order in which to remove the customers
void sort_by_llm_1(std::vector<int>& customers, const Instance& instance) {
    // Placeholder for LLM-based sorting logic
    // This function should implement the logic to sort customers based on a learned
     model
    // For now, we will just sort randomly as a placeholder
    // sort_randomly(customers, instance);
    static thread_local std::mt19937 gen(std::random_device{}());
    std::shuffle(customers.begin(), customers.end(), gen);
}
```

Prompt 10: Seed heuristic ({seed_code}).

# B ADDITIONAL DETAILS

## B.1 COSTS PER RUN

Table 2: Comparison token usage and cost estimates across models per run (as of Sep. 2025) with inference providers sources.

| Model | Open Source | Token Usage | | Costs ($) | | Total Costs ($) | Source |
|---|---|---|---|---|---|---|---|
| | | Input | Output | Input | Output | | |
| Gemini 2.0 Flash | ✗ | 2.5M | 1.2M | 0.10 | 0.40 | 0.73 | Vertex AI |
| Gemini 2.5 Flash | ✗ | 4.1M | 7.1M | 0.30 | 2.50 | 18.98 | Vertex AI |
| gpt-oss (120B) | ✓ | 4.0M | 2.5M | 0.09 | 0.36 | 1.26 | Clarifai |
| gemma 3 (27B) | ✓ | 2.5M | 1.2M | 0.09 | 0.16 | 0.42 | DeepInfra |
| Qwen3 (30B) | ✓ | 1.5M | 4.4M | 0.08 | 0.29 | 1.40 | Clarifai |
| Llama 3.3 (70B) | ✓ | 2.3M | 1.0M | 0.08 | 0.29 | 0.47 | Clarifai |

## B.2 USE OF LARGE LANGUAGE MODELS

LLMs played an active role in this work. Beyond serving as general-purpose writing assistants for improving clarity, style, and grammar and as coding assistants, LLMs were employed as heuristic discovery tools during the optimization phase of our study. Importantly, the core research contributions, including the design of the framework, theoretical development, and validation of results, were conceived, implemented, and verified exclusively by the authors. All outputs from LLMs were critically assessed, refined, and integrated to ensure correctness and adherence to academic standards.

# C EVALUATION ON CVRPLIB INSTANCES

We also evaluate VRPAGENT on the X instance set (Uchoa et al., 2017) from CVRPLib, which consists of 100 CVRP instances with sizes ranging from 100 to 1000 customers. This dataset is ex-

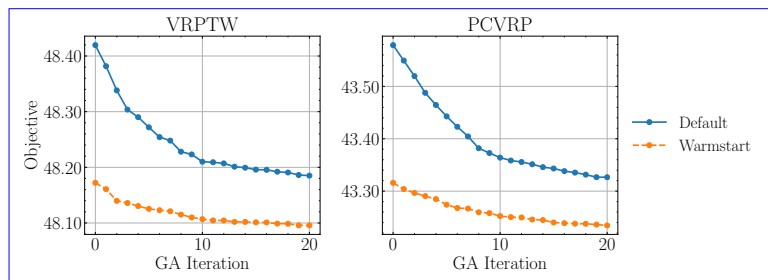

Figure 6: Objective values during heuristic discovery for VRPTW and PCVRP with and without warm-starts from the best CVRP heuristic.

plicitly designed to encompass a wide variety of challenging problem structures, including instances with highly clustered customers and heterogeneous demands, providing a rigorous test of our agent's robustness in scenarios that closely resemble real-world complexity. To generate a heuristic for these instances, we train VRPAGENT on a separate set of 128 instances exhibiting similar characteristics. During the heuristic discovery process, the runtime for each instance is constrained to $0.025n$ seconds, where $n$ denotes the number of nodes.

The resulting heuristic is subsequently evaluated on the full X instance set with an increased runtime limit of $1.0n$ seconds per instance. We compare the performance of the heuristic generated by VR-PAGENT against the state-of-the-art handcrafted method SISRs under the same runtime constraints. For each instance, we perform 10 independent runs and report the average results.

Table 3 summarizes the performance of both approaches on all individual instances. VRPAGENT achieves superior solutions compared to SISRs on 67 instances, whereas SISRs outperforms VRPA-GENT on 24 instances, demonstrating the overall advantage of our approach in this setting. Notably, for larger instances with more than 500 customers, the heuristic produced by VRPAGENT consistently yields better solutions, with only a few exceptions, highlighting its effectiveness in scaling to more complex and larger-scale problems.

## D  SUCCESS RATE

During the main training runs, we track the proportion of generated operators that function correctly, i.e., those that both compile successfully and produce solutions for all validation instances. We refer to this proportion as the success rate, and it provides insight into the difficulty of the generation tasks posed to the LLMs.

Fig. 7 reports the success rate over the course of the discovery runs. Across all problem settings, the success rate remains above 85%, indicating that the generation tasks are generally manageable for the LLM used. For the VRPTW, we also observe a gradual increase in success as the discovery process progresses. For the PCVRP, the success rate is consistently around 95%, likely because not all customers must be visited, making feasible solution generation comparatively easy (e.g., even a solution with no tours is technically feasible).

Note that the figure reflects the success rate only for operators generated within the main loop of VRPAGENT-GA (i.e., offspring and mutation operators). During the initial population generation, the success rates are lower: 73% for the CVRP, 68% for the VRPTW, and 90% for the PCVRP.

## E  WARM-STARTING HEURISTIC DISCOVERY

We investigate whether operators discovered for one problem can be effectively adapted to other problems by conducting experiments in which we warm-start the discovery process for the VRPTW and PCVRP using the best heuristic found for the CVRP. For the warm-start, we provide the CVRP implementation as an example to the model and task it with adapting the heuristic to the new problem (i.e., VRPTW or PCVRP). Each configuration is run 10 times, and the results are averaged.

Table 3: Comparisons on the X set of CVRPLib: best-known solutions (BKS), SISRs and VRPAgent costs, and their percentage gaps. Results show the average performance across 10 runs.

| Instance | BKS | SISRs Obj. | Gap% | VRPAgent Obj. | Gap% | Instance | BKS | SISRs Obj. | Gap% | VRPAgent Obj. | Gap% |
|---|---|---|---|---|---|---|---|---|---|---|---|
| X-n101-k25 | 27591 | 27591 | 0.000 | 27591 | 0.000 | X-n336-k84 | 139111 | 139515 | 0.290 | **139377** | **0.190** |
| X-n106-k14 | 26362 | 26383 | 0.080 | **26366** | **0.020** | X-n344-k43 | 42050 | **42110** | **0.140** | 42146 | 0.230 |
| X-n110-k13 | 14971 | 14971 | 0.000 | 14971 | 0.000 | X-n351-k40 | 25896 | **25982** | **0.340** | 25992 | 0.370 |
| X-n115-k10 | 12747 | 12747 | 0.000 | 12747 | 0.000 | X-n359-k29 | 51505 | 51575 | 0.140 | **51570** | **0.130** |
| X-n120-k6 | 13332 | **13332** | **0.000** | 13332 | 0.010 | X-n367-k17 | 22814 | 22866 | 0.230 | **22829** | **0.070** |
| X-n125-k30 | 55539 | 55606 | 0.120 | **55545** | **0.010** | X-n376-k94 | 147713 | 147814 | 0.070 | **147737** | **0.010** |
| X-n129-k18 | 28940 | **28952** | **0.040** | 28955 | 0.050 | X-n384-k52 | 65928 | 66128 | 0.300 | **66066** | **0.210** |
| X-n134-k13 | 10916 | 10943 | 0.250 | **10942** | **0.240** | X-n393-k38 | 38260 | 38393 | 0.350 | **38355** | **0.250** |
| X-n139-k10 | 13590 | 13599 | 0.070 | **13596** | **0.050** | X-n401-k29 | 66154 | 66258 | 0.160 | **66251** | **0.150** |
| X-n143-k7 | 15700 | 15717 | 0.110 | 15716 | 0.110 | X-n411-k19 | 19712 | 19785 | 0.380 | **19764** | **0.270** |
| X-n148-k46 | 43448 | **43478** | **0.070** | 43492 | 0.100 | X-n420-k130 | 107798 | 107891 | 0.090 | 107897 | 0.090 |
| X-n153-k22 | 21220 | **21227** | **0.030** | 21316 | 0.460 | X-n429-k61 | 65449 | 65609 | 0.240 | **65590** | **0.220** |
| X-n157-k13 | 16876 | 16883 | 0.040 | **16876** | **0.000** | X-n439-k37 | 36391 | 36485 | 0.260 | **36461** | **0.190** |
| X-n162-k11 | 14138 | **14153** | **0.110** | 14153 | 0.120 | X-n449-k29 | 55233 | **55420** | **0.340** | 55443 | 0.380 |
| X-n167-k10 | 20557 | **20557** | **0.000** | 20590 | 0.160 | X-n459-k26 | 24139 | 24247 | 0.450 | **24226** | **0.360** |
| X-n172-k51 | 45607 | 45630 | 0.050 | **45607** | **0.000** | X-n469-k138 | 221824 | 222339 | 0.230 | **222249** | **0.190** |
| X-n176-k26 | 47812 | **47841** | **0.060** | 47889 | 0.160 | X-n480-k70 | 89449 | **89568** | **0.130** | 89603 | 0.170 |
| X-n181-k23 | 25569 | 25580 | 0.040 | **25576** | **0.030** | X-n491-k59 | 66483 | 66708 | 0.340 | **66608** | **0.190** |
| X-n186-k15 | 24145 | **24163** | **0.080** | 24180 | 0.140 | X-n502-k39 | 69226 | 69275 | 0.070 | 69272 | 0.070 |
| X-n190-k8 | 16980 | 16996 | 0.090 | **16990** | **0.060** | X-n513-k21 | 24201 | **24295** | **0.390** | 24298 | 0.400 |
| X-n195-k51 | 44225 | 44308 | 0.190 | **44277** | **0.120** | X-n524-k153 | 154593 | 154905 | 0.200 | **154786** | **0.120** |
| X-n200-k36 | 58578 | 58641 | 0.110 | **58635** | **0.100** | X-n536-k96 | 94846 | 95209 | 0.380 | **95153** | **0.330** |
| X-n204-k19 | 19565 | **19631** | **0.340** | 19675 | 0.570 | X-n548-k50 | 86700 | 86806 | 0.120 | **86783** | **0.100** |
| X-n209-k16 | 30656 | **30670** | **0.050** | 30679 | 0.080 | X-n561-k42 | 42717 | 42878 | 0.380 | **42820** | **0.240** |
| X-n214-k11 | 10856 | 10904 | 0.450 | **10897** | **0.380** | X-n573-k30 | 50673 | 50826 | 0.300 | **50773** | **0.200** |
| X-n219-k73 | 117595 | 117624 | 0.030 | **117612** | **0.020** | X-n586-k159 | 190316 | 190699 | 0.200 | **190598** | **0.150** |
| X-n223-k34 | 40437 | 40551 | 0.280 | **40491** | **0.140** | X-n599-k92 | 108451 | 108706 | 0.240 | **108685** | **0.220** |
| X-n228-k23 | 25742 | **25794** | **0.200** | 25797 | 0.220 | X-n613-k62 | 59535 | 59765 | 0.390 | **59679** | **0.240** |
| X-n233-k16 | 19230 | 19282 | 0.270 | 19281 | 0.270 | X-n627-k43 | 62164 | 62335 | 0.280 | **62320** | **0.250** |
| X-n237-k14 | 27042 | **27103** | **0.230** | 27134 | 0.340 | X-n641-k35 | 63682 | 63877 | 0.310 | **63842** | **0.250** |
| X-n242-k48 | 82751 | 82902 | 0.180 | **82879** | **0.150** | X-n655-k131 | 106780 | 106884 | 0.100 | **106825** | **0.040** |
| X-n247-k50 | 37274 | 37389 | 0.310 | **37342** | **0.180** | X-n670-k130 | 146332 | **146992** | **0.450** | 147491 | 0.790 |
| X-n251-k28 | 38684 | 38808 | 0.320 | **38737** | **0.140** | X-n685-k75 | 68205 | 68401 | 0.290 | **68340** | **0.200** |
| X-n256-k16 | 18839 | 18901 | 0.330 | **18885** | **0.250** | X-n701-k44 | 81923 | 82131 | 0.260 | **82080** | **0.190** |
| X-n261-k13 | 26558 | 26657 | 0.380 | **26650** | **0.350** | X-n716-k35 | 43373 | **43491** | **0.270** | 43493 | 0.280 |
| X-n266-k58 | 75478 | 75650 | 0.230 | 75651 | 0.230 | X-n733-k159 | 136187 | 136462 | 0.200 | **136401** | **0.160** |
| X-n270-k35 | 35291 | **35347** | **0.160** | 35362 | 0.200 | X-n749-k98 | 77269 | 77584 | 0.410 | **77493** | **0.290** |
| X-n275-k28 | 21245 | **21275** | **0.150** | 21285 | 0.190 | X-n766-k71 | 114417 | 114761 | 0.300 | **114688** | **0.240** |
| X-n280-k17 | 33503 | 33634 | 0.390 | **33559** | **0.170** | X-n783-k48 | 72386 | 72645 | 0.360 | **72568** | **0.250** |
| X-n284-k15 | 20215 | 20289 | 0.370 | **20272** | **0.280** | X-n801-k40 | 73305 | 73446 | 0.190 | **73438** | **0.180** |
| X-n289-k60 | 95151 | 95452 | 0.320 | **95421** | **0.280** | X-n819-k171 | 158121 | 158513 | 0.250 | **158408** | **0.180** |
| X-n294-k50 | 47161 | 47280 | 0.250 | **47268** | **0.230** | X-n837-k142 | 193737 | 194047 | 0.160 | **194018** | **0.150** |
| X-n298-k31 | 34231 | 34274 | 0.120 | **34264** | **0.100** | X-n856-k95 | 88965 | 89152 | 0.210 | **89119** | **0.180** |
| X-n303-k21 | 21736 | **21788** | **0.240** | 21819 | 0.380 | X-n876-k59 | 99299 | 99544 | 0.250 | **99494** | **0.200** |
| X-n308-k13 | 25859 | 26192 | 1.290 | **25934** | **0.290** | X-n895-k37 | 53860 | 54138 | 0.520 | **54104** | **0.450** |
| X-n313-k71 | 94043 | 94235 | 0.200 | **94189** | **0.160** | X-n916-k207 | 329179 | **329556** | **0.110** | 329566 | 0.120 |
| X-n317-k53 | 78355 | 78397 | 0.060 | **78364** | **0.010** | X-n936-k151 | 132715 | 133387 | 0.510 | **133201** | **0.370** |
| X-n322-k28 | 29834 | 29933 | 0.330 | **29905** | **0.240** | X-n957-k87 | 85465 | 85640 | 0.200 | 85633 | 0.200 |
| X-n327-k20 | 27532 | 27671 | 0.510 | **27636** | **0.380** | X-n979-k58 | 118976 | 119147 | 0.140 | **119118** | **0.120** |
| X-n331-k15 | 31102 | **31141** | **0.130** | 31164 | 0.200 | X-n1001-k43 | 72355 | 72563 | 0.290 | **72536** | **0.250** |
| | | | | | | Avg. Gap% | * | 0.229 % | | **0.195%** | |

Fig. 6 shows the objective function values over the course of the discovery process for the VRPTW and PCVRP, comparing warm-start and cold-start runs. Warm-starting significantly improves performance, yielding better results from the initial population. All warm-start runs begin from the best CVRP implementation found across 10 discovery runs, which required a substantial amount of compute. The combination of high-quality initial heuristics and diversity introduced through the start population generation allows the method to discover implementations that outperform the cold-start runs on average.

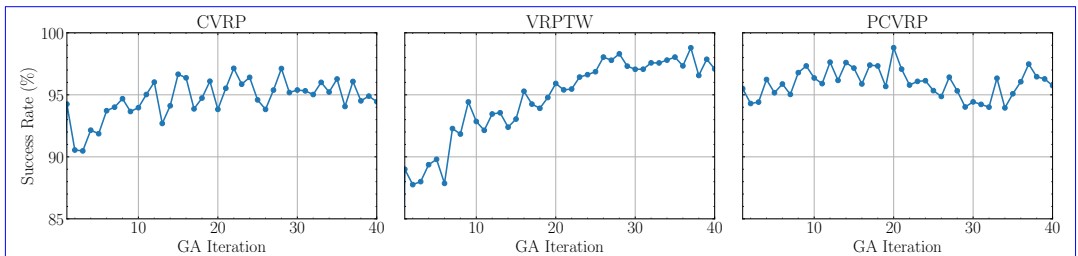

Figure 7: Percentage of operators that compile successfully and produce valid solutions (success rate) throughout the main discovery runs.

# F  ANALYSIS OF DISCOVERED HEURISTICS

## F.1  HIGH-LEVEL DESCRIPTION

### F.1.1  CVRP

**Customer Removal**  The removal heuristic, iteratively constructs a set of customers to remove based on a target size drawn from a uniform distribution (e.g., between 15 and 28). The process initializes by either cutting a random contiguous tour segment (with probability 0.2) or selecting a single random seed. The set is subsequently expanded by choosing a pivot from the currently selected customers and applying one of three weighted strategies: adding a spatial neighbor from the pivot's k-nearest graph (60%), adding a short contiguous segment from the pivot's current tour (25%), or adding the pivot's immediate tour predecessor or successor (15%). A random fallback ensures the target removal count is always met.

**Customer Ordering**  The customer ordering operator stochastically selects one of eight sorting rules and a random sorting direction (ascending or descending). The scoring criteria include geometric metrics (distance to depot, distance to the removed set's centroid, polar angle), problem-specific attributes (demand, combined demand-distance), and topological features (connectivity among removed nodes). Alternatively, the order may be determined by a stochastic Nearest-Neighbor chain or a simple random shuffle. To enhance diversity, minor additive noise is applied to the calculated scores during the sorting process.

### F.1.2  VRPTW

**Customer Removal**  The removal operator, constructs a cluster of customers to remove by growing a set from an initial random seed up to a target size drawn uniformly between 10 and 15. The expansion logic prioritizes connectivity: at each step, the algorithm attempts to select a tour neighbor (immediate predecessor or successor in the current route) with a probability of 0.45, or a geographical neighbor from the instance's adjacency list with a probability of 0.35, where the latter is sampled with a power-law bias to favor closer nodes. If these connected expansion attempts fail, or with a 15% probability once the set is sufficiently large, the algorithm performs a "random jump" or falls back to a uniform random selection to ensure the target removal count is met.

**Customer Ordering**  The customer ordering operator uniformly selects one of ten available scoring strategies. These strategies range from sorting by single raw attributes (e.g., time window width, start time, or negative demand) to complex composite scores. Specifically, one advanced strategy aggregates five normalized attributes (including service time and distance to depot) using randomly generated weights and polarities, while another assesses demand density relative to time window tightness. To maximize diversity, the final ordering process includes a microscopic noise factor for tie-breaking, a 25% probability of reversing the sorted list, and a final pass that applies a small number of random swaps.

### F.1.3  PCVRP

**Customer Removal**   The removal operator, constructs a cluster of customers to remove by growing a set from an initial random seed up to a target size drawn uniformly between 10 and 20. The seeding process prioritizes unassigned customers (25%) and those currently belonging to tours (60%) before falling back to a uniform selection. The expansion logic is iterative: at each step, a subset of "source" customers is chosen from the current selection, and candidates are gathered from both their topological neighbors (via the adjacency list) and random samples of their current tour-mates. A single candidate is uniformly selected from this pool, with a random safeguard applied if the candidate pool is empty.

**Customer Ordering**   The customer ordering operator executes a pure random shuffle with probability 0.1; otherwise, it employs a score-based sort relative to a pivot node. The pivot is selected from the removed set (80%) or set to the depot (20%). The scoring function calculates a weighted sum involving the distance to the pivot, the distance to the depot, the customer's demand, and their prize (if applicable). To enforce diversity, the specific weights for these components are modulated by three distinct parameter regimes—varying the emphasis between prize collection and distance minimization—and the final list is sorted in either ascending or descending order with equal probability.

### F.2  EXPERT EVALUATION

LLM generated code raises many questions about its quality and maintainability. A further question is how the code works and how it is able to achieve state-of-the-art performance. While we are unable to fully answer these questions, we try to provide some initial insights into the quality of the best heuristic generated for each problem. To do this, we have three co-authors of the paper with many years of experience writing heuristics by hand analyze the code according to several criteria. We acknowledge that this is not a scientific study and is not intended to draw generalizations about the ability of LLMs to code heuristics for optimization problems. Rather, our goal is to give some indications as to how the code generated compares to code written by humans and what kind of ideas are present. We note that the code is generated without comments to avoid the LLM influencing the analysis, however we note that variable names are present that do give some contextual information about what the code does.

The three heuristics experts have XX (expert 1), YY (expert 2) and ZZ (expert 3) years of experience writing OR heuristics[1]. All experts have experience with routing problems in addition to other types of OR problems. Each expert provides an evaluation of the generated code of the best performing heuristic for each of the three problems examined in this work. The experts describe a consensus description of how the heuristic works then write independent discussions of each heuristic. The individual rating criteria are as follows:

1. Readability (noting that the assessments are not general statements about LLM code)
2. Coherence and soundness
3. Maintainability
4. Interpretability, i.e., do we know why this code works well?
5. Are there any new ideas in the heuristic?

### F.2.1  CVRP

The removal and sorting mechanisms are best described as ensembles of heuristics in which the heuristic applied at any given iteration is chosen at random according to a probability distribution determined through the static parameters of the approach. For the selection of customers for removal, the heuristics of the ensemble show a similarity to the SISRs heuristic. In the first, adjacent customers are selected for removal and in the second, random segments of tours are chosen. Since

---

[1]To avoid potentially violating the double blind submission policy, we do not indicate the years of experience of the experts, as they are all coauthors of the work. These will be provided in the accepted version of this work, and this message will be removed.

these segments can overlap, we also have a SISRs-like idea. The third heuristic, as best as we can determine, tries to expand a tour segment. For sorting, the heuristic first decides whether to sort descending or ascending according to one of seven different heuristics. We omit a detailed description of all the heuristics, but note that these include generally known ideas for sorting customers in a CVRP, e.g. using criteria such as the distance to the depot, the demand of the customers, weighted combinations of distance and demand, and greedy nearest-neighbor sequencing.

**Expert 1**  The code is surprisingly well-written and is split into different functions in a logical fashion. The heuristics are coherent and reasonable. The ensemble contains many components and an ablation analysis would be necessary to determine how much each component actually contributes to the solution quality. The code looks relatively easy to maintain and follows good design principles for C++ code. There are no special constructs or libraries used, the memory management is very simple. While the heuristic does not present anything radically new, the scoring mechanism for removed neighbor connectivity seems looks novel. The removal heuristic is a SISRs variant, but not exactly the same as the original.

**Expert 2**  The code is easy to understand and well-written. It contains both very meaningful high-level comments that make it easy to quickly assess the big picture of main functions, and low-level comments that facilitate understanding the details. In addition, the code is decomposed in a useful way into smaller functions with good names. As far as I see, this is good and standard C++ code that should be very maintainable and does not contain any weird parts. The created selection heuristics can be characterized as an ensemble of various known (and reasonably different) approaches that are applied in a random fashion. The sorting function randomly chooses between eight scoring schemes that involve use meaningful criteria; I assume that some combinations of criteria used for scoring are novel, although the criteria themselves are mostly not.

**Expert 3**  The code is generally well-structured, with its components split into separate functions. This organization makes it easier for users to follow the logic without becoming overwhelmed by a single large code block. Additionally, the descriptive variable names and comments help readers quickly recognize the purpose of each section and provide useful guidance when deeper understanding is needed. Conceptually, the code combines several known ideas from the literature in a structured ensemble; while none of the components are new by themselves, the way they are combined is somewhat novel. However, because it is not immediately clear which components are essential in practice, further analysis would be required to determine whether the code could be simplified.

### F.2.2  VRPTW

The selection and sorting heuristics form portfolio approaches, that is, ensembles of different heuristics that are probabilistically selected and combined. The selection heuristic removes a randomly chosen number of customers (10-15). It starts with a randomly picked customer and then uses a main loop in which one additional customer is added per iteration, using spatial proximity and tour neighborhood to a selected reference customer as well as pure randomness to guide the choice. The sorting code involves ten different strategies, of which one is randomly chosen per call of the heuristic. Five strategies rely on simple smetrics such as demand or time window start time, four are based on combined metrics and one is a purely random shuffling of the customers. One of the combined metrics is highly complex and heavily parameterized.

**Expert 1**  This heuristic is again very easy to read and the code is nicely structured. The functions are decently easy to understand and the code can be considered interpretable. While the sorting criteria are mostly standard and there are no big surprises, some combinations of metrics (e.g., the normalized combination in case 7) are potentially novel.

**Expert 2**  Like in the CVRP case, the code is very well explained in both high and low-level comments and structured into functions in a meaningful way, avoiding deeply nested statements and making the code easy to follow and maintain. The code is reasonable and concise modern C++. The selection heuristic randomly applies a set of reasonable heuristics, and the sorting code again is based on randomly choosing between a set of useful and known sorting criteria, some of which are weighted combinations of multiple criteria. Although consisting of simple components,

the big number of heuristics and the multitude of parameters are remarkable. A human expert may have come up with similar ideas, but selecting the components and tuing the parameters would have taken a lot of time.

**Expert 3**   The code is overall readable and, as in the other case, reflects a consistent style the LLM uses. Splitting the implementation into several functions helps users understand the individual components more easily, while the comments and variable names further enhance readability. Again, the sorting section includes many variants, but the large number of options risks obscuring which criteria actually matter. Overall, the components used are standard for the VRPTW.

### F.2.3   PCVRP

The removal and sorting heuristics form an ensemble of heuristics, with the specific method chosen at random according to static parameters defined at the beginning. The removal operator first determines how many customers to remove. It starts the selection from either an unvisited customer, a customer based on a random tour, or a purely random choice. Customers are then added iteratively based on adjacency and tour neighbors, with random selection used as a fallback. The sorting operator either shuffles customers randomly or scores them using a combination of properties such as prize, distance to the depot, and demand, with probabilities applied to vary the weights of these properties.

**Expert 1**   The code is well structured as with the other problems. A weakness in this code is one code block in which many "magic numbers" are inserted into the code along with many random numbers. While this code is not hard to understand, it could be written in a cleaner way. The scoring mechanism is likely novel.

**Expert 2**   The code is mostly well-written and involves mostly meaningful explanations both in high-and low-level comments. Again, it is reasonably structured into simple and mostly easy-to-understand functions. Both the selection and sorting heuristics seem reasonable and involve mostly known component heuristics and criteria. Compared the CVRP and the VRPTW, the LLM-generated sorting function involves only two main strategies, one of which is a weighted combination where the weights are selected using multiple nested random number draws. This is a bit hard to make sense of, I am sure that a similar behavior could have been achieved in an easier-to understand way (e.g. by framing the highest level of the nested random selection as separate strategies instead of introducing what the LLM calls "sub-biases" in a single strategy).

**Expert 3**   The code is well-structured and generally readable, though some functions are longer than in the previous heuristic and combine multiple components that could be separated into distinct functions, as was done in the CVRP and VRPTW implementations. Nevertheless, the code remains understandable thanks to the accompanying comments. While the components themselves are familiar and standard for this problem domain, some of their combinations are unusual.

### F.3   ABLATION STUDY

We conduct ablation studies for the best-performing heuristics to better understand their internal mechanisms. To this end, we systematically remove individual subcomponents from the designed heuristics and assess the impact of each change. To ensure statistically meaningful results, we evaluate all variants on larger validation sets comprising 500 instances per problem, using the same 60s per-instance runtime limit as in the main experiments. For every configuration, including the default version (which contains none of the ablated modifications), we run VRPAGENT 10 times on the same set of 500 test instances and record the average tour cost of each run. We report the mean and standard deviation of these run-level averages, together with a 95% confidence interval. Statistical significance relative to the default configuration is determined using the Wilcoxon signed-rank test applied to the paired run averages.

### F.3.1   CVRP

**Customer Removal**   We ablate the following subcomponents that are used by the generated heuristic during the customer removal procedure:

1. **Initial Segment Removal:** This mechanism is employed as an alternative to selecting a single seed customer. It selects a fixed-length contiguous segment of customers for removal to initiate the destruction process. In the subsequent expansion steps, this tour segment is then extended by other customers.

2. **Nearest Neighbor Expansion:** Used in the iterative removal loop, this mechanism grows the removed set by spatial clustering around already-selected customers. It selects a candidate $c$ from the $k$-nearest neighbors of a pivot $p$ using a biased probability distribution, which strongly favors the nearest physical neighbor; this is intended to ensure the final removed set is geographically compact, creating a large, localized "hole".

3. **Adjacent Tour Node Expansion:** This alternative expansion strategy works by identifying the immediate predecessor and successor of a pivot customer $p$ in its current tour and then uniformly selecting one of these adjacent (unselected) nodes. Its intention is to ensure the removal of customers that are tightly coupled in the solution tour.

4. **Route Segment Removal:** This alternative expansion strategy works by identifying a segment of fixed length that includes the pivot $p$ in its tour and adding all nodes in that segment to the removed set.

Table 4 shows the results of the ablation experiments. Removing Route Segment Removal significantly or Nearest Neighbor Expansion leads to a significant increase in costs. Other components, such as Initial Segment Removal and Adjacent Tour Node Expansion, have only minor effects.

Table 4: Ablation of removal strategies for the CVRP.

| Configuration | Mean | Std | CI95 | p-value |
|---|---|---|---|---|
| Default Configuration | 36.70378 | 0.003310 | 0.002368 | — |
|   - w/o Initial Segment Removal | 36.70551 | 0.003138 | 0.002245 | 0.375000 |
|   - w/o Adjacent Tour Node Expansion | 36.70705 | 0.002539 | 0.001816 | 0.064453 |
|   - w/o Route Segment Removal | 36.71770 | 0.003249 | 0.002324 | 0.001953 |
|   - w/o Nearest Neighbor Expansion | 37.72012 | 0.018678 | 0.013362 | 0.001953 |

**Customer Ordering** We ablate the following subcomponents that are used by the generated heuristic during the customer ordering procedure:

1. **Demand:** A simple ordering rule that orders removed customers based on the customer's demand value in descending order, ensuring high-demand customers are prioritized early during the re-insertion phase.

2. **Center Proximity:** This ordering rule first calculates the geometric center (centroid) of all removed customers, and then sorts them by their Euclidean distance from this centroid.

3. **Depot Proximity:** This ordering rule calculates the score as the raw Euclidean distance from the depot. The list is typically sorted in ascending order of this distance.

4. **Polar Angle:** This rule defines a radial sequencing of insertion priorities. The mechanism calculates the polar angle of each customer relative to the depot (origin) and then sorts the list ascending or descending by this angle to implement a sweep re-insertion strategy.

5. **Connectivity:** This rule prioritizes customers based on their spatial integration with the removed set. The mechanism calculates the score by measuring the connectivity, determined by the sum of inverse distances between each customer and all other customers in the removed set; this is intended to prioritize the re-insertion of customers that are highly integrated with the destroyed cluster.

6. **Weighted Demand-Distance Score:** This rule prioritizes customers based on a weighted sum of demand and distance to the depot. The mechanism calculates a linear combination score, where the weights are varied stochastically and a small noise value is applied to increase exploration.

7. **Nearest Neighbor Chain:** This rule defines a local, greedy sequence. It starts with a random customer and then iteratively finds the **closest unplaced customer** to the most

recently selected one until the entire set is ordered; this generates a local, greedy sequence for the destroyed cluster.

8. **Random Shuffle:** This rule simply orders all removed customers at random.

Table 5 shows the results. For most ordering rules, we do not observe a statistically significant impact, which is expected given that many strategies overlap and the effect of each individual rule is limited when numerous strategies are combined. Interestingly, the Connectivity rule stands out as a significant contributor to performance, despite being more computationally expensive than most other strategies.

Table 5: Ablation of ordering strategies for the CVRP.

| Configuration | Mean | Std | CI95 | p-value |
|---|---|---|---|---|
| Default Configuration | 36.70378 | 0.003310 | 0.002368 | — |
| - w/o Nearest Neighbor Chain | 36.70083 | 0.001283 | 0.000918 | 0.037109 |
| - w/o Demand | 36.70116 | 0.003045 | 0.002178 | 0.130859 |
| - w/o Weighted Demand-Distance Score | 36.70196 | 0.001920 | 0.001374 | 0.173071 |
| - w/o Random Shuffle | 36.70390 | 0.002159 | 0.001545 | 0.625000 |
| - w/o Polar Angle | 36.70499 | 0.001673 | 0.001197 | 0.431641 |
| - w/o Connectivity | 36.70747 | 0.001244 | 0.000890 | 0.009766 |
| - w/o Center Proximity | 36.70984 | 0.003388 | 0.002424 | 0.003906 |
| - w/o Depot Proximity | 36.75184 | 0.002093 | 0.001498 | 0.001953 |

### F.3.2 VRPTW

**Customer Removal** We ablate the following subcomponents that are used by the generated heuristic during the customer removal procedure:

1. **Tour Neighbor Expansion:** This mechanism expands the removed set by identifying the immediate predecessor or successor of a pivot customer in its current tour. It is intended to remove chains of customers that are in the same solution tour.

2. **Nearest Neighbor Expansion:** Used to maintain spatial locality, this mechanism selects a candidate from the closest customers of a pivot customer. It utilizes a biased probability distribution that strongly favors the nearest neighbors to ensure the destroyed set remains geographically compact.

3. **Random Jump Expansion:** This mechanism introduces a perturbation to the growth process by selecting a completely random unselected customer, regardless of proximity to the current set. It is only triggered once the removed set reaches a minimum size.

Table 6 presents the results. Removing the Random Jump Expansion has no statistically significant effect on the VRPTW heuristic, while both of the other expansion strategies are critical for maintaining high solution quality.

Table 6: Ablation of selection strategies for the VRPTW.

| Configuration | Mean | Std | CI95 | p-value |
|---|---|---|---|---|
| Default Configuration | 47.46744 | 0.005987 | 0.004283 | — |
| - w/o Random Jump Expansion | 47.46464 | 0.004139 | 0.002961 | 0.375000 |
| - w/o Tour Neighbor Expansion | 47.61712 | 0.003621 | 0.002590 | 0.001953 |
| - w/o Nearest Neighbor Expansion | 48.16588 | 0.005831 | 0.004171 | 0.001953 |

**Customer Ordering** We ablate the following subcomponents that are used by the generated heuristic during the customer ordering procedure:

1. **Time Window Width:** This rule orders removed customers based on the width of their time window in ascending order, ensuring that customers with tighter time constraints are prioritized for re-insertion.

2. **Time Window Start:** This rule orders customers strictly by the start time of their time window in ascending order.

3. **Demand:** A rule that orders customers based on their demand value in descending order, prioritizing high-demand customers early in the insertion process.

4. **Depot Distance:** This rule orders customers based on their Euclidean distance from the depot in descending order, prioritizing the re-insertion of outlying customers before those closer to the depot.

5. **Weighted Start and Width:** This mechanism calculates a deterministic score as a linear combination of the time window start time and half the time window width; it prioritizes customers that start early and have tight constraints.

6. **Service Time:** This rule orders customers based on their service duration in descending order, prioritizing those that consume the most route time.

7. **Multi-Attribute Score:** This rule prioritizes customers using a linear combination of five normalized attributes: time window start, time window width, demand, service time, and distance. The weights for each attribute are randomized for every execution to maximize exploration of different sorting criteria.

8. **Arrival Time Slack:** This rule prioritizes customers based on the maximum possible slack between the earliest arrival (determined by distance from depot) and the latest possible arrival (determined by the time window end). It orders customers with larger slack values first.

9. **Weighted Density and Tightness:** This rule calculates a composite score combining demand density (demand divided by service time) and time window tightness (inverse of width). It weights these two components using a stochastic factor to alternate priority between high-density and highly-constrained customers.

10. **Pure Random Shuffle:** This rule simply orders all removed customers at random.

Table 7 shows the results. For most ordering strategies, there is no statistical evidence that they individually have a meaningful impact on the heuristic's performance. This is expected, as 10 different strategies are used and selected uniformly at random during the search, so removing a single strategy has only a minor effect. Notably, the Weighted Start and Width heuristic stands out as an exception, significantly contributing to the overall performance.

Table 7: Ablation of ordering strategies for the VRPTW.

| Configuration | Mean | Std | CI95 | p-value |
|---|---|---|---|---|
| Default Configuration | 47.46744 | 0.005987 | 0.004283 | — |
| - w/o Demand | 47.46509 | 0.003528 | 0.002524 | 0.556641 |
| - w/o Time Window Width | 47.46617 | 0.002892 | 0.002069 | 0.431641 |
| - w/o Arrival Time Slack | 47.46631 | 0.006066 | 0.004339 | 0.695312 |
| - w/o Multi-Attribute Score | 47.46645 | 0.006211 | 0.004443 | 1.000000 |
| - w/o Depot Distance | 47.46671 | 0.007461 | 0.005337 | 1.000000 |
| - w/o Time Window Start | 47.46708 | 0.004177 | 0.002988 | 1.000000 |
| - w/o Weighted Density and Tightness | 47.46756 | 0.005045 | 0.003609 | 1.000000 |
| - w/o Pure Random Shuffle | 47.46765 | 0.006718 | 0.004806 | 0.845703 |
| - w/o Service Time | 47.46789 | 0.007746 | 0.005541 | 0.921875 |
| - w/o Weighted Start and Width | 47.49367 | 0.005458 | 0.003905 | 0.001953 |

### F.3.3 PCVRP

**Customer Removal** We ablate the following subcomponents that are used by the generated heuristic during the customer removal procedure:

1. **Biased Seed Selection:** Instead of selecting the initial seed customer uniformly at random, this mechanism prioritizes the selection of customers that are currently visited by a vehicle over unvisited customers.

2. **Multi-Source Expansion:** Used in the iterative removal process, this strategy selects customers using multiple seed customer, allowing the destruction area to expand in clusters.

3. **Tour-Based Neighbor Expansion:** This mechanism expands the removed set by identifying customers that share the same route as a seed customer. It samples a subset of tour-neighbors to ensure that entire segments of existing routes are targeted are removed.

Table 8 shows the results. All three ablated removal strategies contribute to the overall performance of the heuristic. Interestingly, even subtle algorithmic details, such as the Biased Seed Selection strategy, which starts the customer removal process from previously visited customers with higher probability rather than uniformly at random, can produce small but consistent improvements.

Table 8: Ablation of selection strategies for the PCVRP.

| Configuration | Mean | Std | CI95 | p-value |
|---|---|---|---|---|
| Default Configuration | 42.73483 | 0.003547 | 0.002538 | — |
| - w/o Multi-Source Expansion | 42.73821 | 0.001865 | 0.001334 | 0.027344 |
| - w/o Biased Seed Selection | 42.75303 | 0.002260 | 0.001617 | 0.001953 |
| - w/o Tour-Based Neighbor Expansion | 42.97553 | 0.003856 | 0.002758 | 0.001953 |

**Customer Ordering** We ablate the following subcomponents that are used by the generated heuristic during the customer ordering procedure:

1. **Pivot Proximity:** This mechanism orders customers based on their distance to a specific pivot node that is stochastically chosen as either the depot or a random customer from the removed set.

2. **Prize-Focused Scoring:** This configuration corresponds to the first weighting profile. It calculates a weighted sum of the customer's prize and their distance to the depot, assigning significantly higher weight to the prize value. A small noise factor is added to encourage slight exploration, but the primary intention is to prioritize the re-insertion of high-profit customers.

3. **Prize-Distance Scoring:** This configuration corresponds to the second weighting profile. It assigns roughly equal weights to the customer's prize and their distance to the depot. To promote solution diversity a significantly larger noise factor is applied compared to the other profiles.

4. **Distance-Demand Scoring:** This configuration corresponds to the third weighting profile. It shifts the heuristic's focus by heavily weighting the distance to the depot and introducing a weight for customer demand (which is ignored in the other profiles).

5. **Random Shuffle:** This mechanism bypasses the scoring logic entirely and orders all removed customers at random. It is selected with a small probability to increase exploration.

Table 9 shows the results. Interestingly, removing the Prize-Distance Scoring slightly improves performance, making it the only case in our ablation experiments where removal leads to a statistically significant improvement. In contrast, both the Distance-Demand Scoring and Pivot Proximity ordering strategies contribute positively and improve performance in a statistically significant manner.

Table 9: Ablation of ordering strategies PCVRP.

| Configuration | Mean | Std | CI95 | p-value |
|---|---|---|---|---|
| Default Configuration | 42.73483 | 0.003547 | 0.002538 | — |
| - w/o Prize-Distance Scoring | 42.73103 | 0.002071 | 0.001481 | 0.013672 |
| - w/o Prize-Focused Scoring | 42.73353 | 0.002825 | 0.002021 | 0.492188 |
| - w/o Random Shuffle | 42.73397 | 0.001956 | 0.001399 | 0.921875 |
| - w/o Distance-Demand Scoring | 42.74467 | 0.003440 | 0.002461 | 0.003906 |
| - w/o Pivot Proximity | 42.86265 | 0.002567 | 0.001837 | 0.001953 |

