# OpenReview forum: "VRPAgent: LLM-Driven Discovery of Heuristic Operators for Vehicle Routing Problems"
_ICLR.cc/2026/Conference — Submitted to ICLR 2026_

### Official Review · Reviewer_Z5wY · 2025-10-28

**Soundness:** 2
**Presentation:** 2
**Contribution:** 2
**Rating:** 4
**Confidence:** 4

**Summary:**

This paper introduces VRPAGENT, a framework for discovering heuristic operators for Vehicle Routing Problems (VRPs) using large language models (LLMs). The method combines LLM-generated “destroy” and “order” operators with a Large Neighborhood Search (LNS) metaheuristic, leveraging genetic algorithms (GAs) to iteratively evolve improved operators. Although the research motivation and validation results seem feasible, the approach is almost identical to existing LLM-guided heuristic frameworks, which weakens the overall contribution of the paper.

**Strengths:**

1. The approach is clear, and the LLM-guided evolutionary framework has discovered operators that go beyond expert-designed ones.
2. The authors have conducted a certain level of analysis on the generated heuristic operators.

**Weaknesses:**

1.	**Incrementa novelty** — The evolutionary framework of VRPAgent is similar to existing heuristic evolutionary frameworks (such as Heuristics evolution based on LLM, e.g., EoH [1]), and the proposed "code length penalty" is also negligible.
2.	**Empirical overclaiming** — The experimental results of VRPAGENT show only minor improvements compared to existing methods. There is a lack of comparison with LLM-empowered LNS approaches, such as LLM-LNS [2]. It is also unclear how it performs compared to adaptive LNS methods like PPO-ALNS [3].
3.	**Fairness of experiments** — It is not clearly stated whether the comparison methods based on LLM heuristic generation have a similar number of API calls.
4.	**Incomplete experimental analysis** — The study lacks an analysis of aspects such as the convergence of the genetic algorithm or the probability of code correctness.

[1] Evolution of heuristics: Towards efficient automatic algorithm design using large language model. ICML 2024.

[2] Large Language Model-driven Large Neighborhood Search for Large-Scale MILP Problems. ICML 2025.

[3] Reinforcement learning-guided adaptive large neighborhood search for vehicle routing problem with time windows. Journal of Combinatorial Optimization, 2025.

**Questions:**

see weaknesses

---

> ### Author Response · Authors · 2025-11-21
>
> We are grateful for the reviewer's constructive comments and the time taken to evaluate our manuscript. We are encouraged that you found our approach clear and recognized that our LLM-guided framework successfully discovered operators capable of outperforming expert-designed baselines. We also appreciate your acknowledgement of our analysis regarding the generated heuristic operators.
>
> We have carefully addressed your concerns regarding novelty, empirical comparisons, experimental fairness, and the completeness of our analysis below.
>
> ### Weaknesses
>
> **W1: Incremental Novelty**
> We acknowledge that the Genetic Algorithm (GA) used in VRPAgent shares inspiration with concurrent evolutionary frameworks like EoH. However, the GA is merely the optimization engine, not the architectural contribution.
> The primary novelty of VRPAgent lies in the **"Operator-in-the-Loop" framework**. Unlike methods that attempt to generate entire solvers from scratch (which often fails for complex constraints), VRPAgent uses LLMs strictly to generate problem-specific *operators* (destroy and ordering), while embedding them within a robust, high-level Large Neighborhood Search (LNS) metaheuristic.
> This structural decision—placing the "AI agent on a leash"—is what provides the necessary guardrails to ensure correctness and scalability. This design choice is vindicated by our results: VRPAgent outperforms prior methods not because the GA is unique, but because of the overall Operator-in-the-Loop framework.
>
>
> **W2: Empirical Overclaiming and Baselines**
> * **Magnitude of Improvement:** We respectfully note that in the mature field of Vehicle Routing, where heuristics like SISRs have been optimized for years, a gap reduction of **0.30%** is a substantial achievement. In real-world logistics operations at scale, this translates to significant cost and emission reductions.
> * **Comparison to LLM-LNS:** We have updated the paper to include a direct comparison to **LLM-LNS** for the CVRP. Our results demonstrate that VRPAgent significantly outperforms LLM-LNS across all considered instance sizes.
> * **Comparison to PPO-ALNS:** PPO-ALNS is a concurrent work published in late October 2025, after the ICLR submission deadline. While we cannot include it as a mandatory baseline due to the concurrent work policy, we emphasize a key structural difference: PPO-ALNS focuses on smaller instances (up to 100 customers) and requires GPUs. In contrast, VRPAgent is designed for scalability, effectively solving instances with 2,000 customers on a single CPU core. Furthermore, PPO-ALNS produces a black-box policy, whereas VRPAgent generates interpretable, white-box code.
>
>
> **W3: Fairness of Experiments (API Calls)**
> We ensured fairness by standardizing the computational budget across all generation-based methods. Specifically, we limit all methods to the same number of **function evaluations** . Since the evaluation step is the computational bottleneck in discovery, this metric ensures that no method benefits from an unfair amount of "trial and error." The number of function evaluations correlates strongly with the number of API calls, ensuring a level playing field regarding the search budget.
>
>
> **W4: Incomplete Analysis (Convergence and Correctness)**
> We thank the reviewer for highlighting this. We have expanded the manuscript to address these points directly:
> * **Code Correctness:** We have added a new analysis in **Appendix C**. We track the "success rate" (percentage of operators that compile and run successfully). We found that the success rate remains consistently high (above **85%**) across all problems. Interestingly, for the VRPTW, the success rate improves over time as the population evolves, while for the PCVRP it remains near 95% due to the relaxed constraint of not visiting every customer.
> * **Convergence:** We explicitly analyze the convergence behavior of our method in **Figure 4**. This figure plots the performance trajectory over 40 iterations for all three problem variants, demonstrating how the GA effectively improves upon the initial population and eventually surpasses the SOTA handcrafted baselines.

---

### Official Review · Reviewer_fYZd · 2025-10-28

**Soundness:** 2
**Presentation:** 2
**Contribution:** 2
**Rating:** 4
**Confidence:** 4

**Summary:**

This paper proposes a framework for automated heuristic discovery in VRPs using LLMs called VRPAgent. VRPAgent integrates LLM-generated problem-specific operators within a Large Neighborhood Search (LNS) metaheuristic and refines them through a genetic algorithm that employs elitism, biased crossover, and code-length penalty mechanisms.​

Key features include generating problem-specific destroy and insert heuristics via LLMs, and evolving these operators over multiple generations to maximize solution quality while controlling code complexity. The method is evaluated across standard VRPs (capacitated, time windows, prize-collecting), consistently discovering heuristics that outperform handcrafted and previous LLM/learning-based methods on large benchmark instances using only CPU resources.​

The approach offers interpretability, practical efficiency, and a reproducible pipeline for discovering and improving heuristics for combinatorial optimization, highlighting a new path for LLM-driven algorithmic design in operations research.​

The contributions include:
1. A hybrid metaheuristic framework (LLM-in-the-loop LNS) for VRPs where LLMs generate, mutate, and combine code for local operators.
2. A genetic algorithm with code-length penalties to evolve and select the best LLM-generated operators.
3. Demonstrating state-of-the-art or superior performance compared to both expert-designed heuristic solvers and recent neural/LLM solutions on several large VRP benchmarks, with superior interpretability and scalability

**Strengths:**

1. The proposed VRPAgent leverages LLMs to generate and evolve problem-specific destroy/insert heuristics for VRPs, significantly reducing the need for expert-written code and enabling discovery of novel strategies.​

2. The framework combines LLM-generated operators with a genetic algorithm using elitism, biased crossover, and code-length penalties, leading to efficient search and interpretable code that is competitive with and sometimes superior to handcrafted heuristics.​

3. VRPAgent consistently outperforms or matches best-in-class expert and neural approaches on benchmark VRPs (CVRP, VRPTW, PCVRP), working efficiently on CPU and scaling to large instances.​

4. The approach maintains strong performance across multiple VRP classes without requiring expensive hardware. By modularizing heuristic generation and search refinement, it allows for adaptation to different problem types and operator ensembles.​

5. By focusing LLM synthesis on manageable code components within a robust metaheuristic shell, VRPAgent strikes a balance between automated innovation and guarantees of feasibility and quality

**Weaknesses:**

1. Many LLM-generated heuristics discovered by VRPAgent are ensembles or recombinations of standard strategies from the literature (e.g., SISRs-like removal, weighted greedy criteria for sorting). The framework excels at combining known components but provides little evidence of discovering fundamentally new algorithms that would advance state-of-the-art theory for VRPs.​

2. While the code is readable to experts, it is often overly redundant, deeply nested, and filled with hard-to-tune random parameters and magic numbers. Several domain experts in the study noted that a human would write more succinct, interpretable, and maintainable code. The logic behind some probabilistic choices is especially convoluted, making ablation studies and performance analysis difficult.​

3. The performance improvements are attributed to complex ensembles and parameterized strategies, but the paper lacks detailed ablation studies pinpointing which components are truly responsible for gains. This makes it hard to generalize findings beyond the benchmarked VRP instances.​

4. Many key parameters affecting the algorithms' behavior are scattered, sometimes hard-coded and sometimes embedded within random logic, increasing risk of inadvertent misconfiguration. This could hinder code modification, adaptation, or debugging in practice.​

5. The experiments focus on classic VRPs and well-known benchmark formats. There is no evaluation of the heuristics' robustness under noisy, dynamic, or highly custom problem constraints, which are common in operational logistics scenarios.​

6. Although some level of interpretability is claimed, true transparency into how and why the LLM-generated code behaves well is lacking. For many users and practitioners, relying on black-box or stochastic mixtures of heuristics without clear guidance or analysis may be risky.​

7. The framework does not address risks inherent in LLM-generated code, such as silent propagation of bugs, accidental feasibility violations, or malicious prompt engineering in operational settings. This could be critical for industrial deployment.​

8. Claims of extensibility to other combinatorial domains (packing, scheduling) are made, but without any experimental or theoretical evidence.

**Questions:**

1. How sensitive is the genetic search to the initialization of random heuristics and the specific code-length penalty functions? Can you provide detailed ablation studies quantifying how different parameter choices impact final solution quality and code interpretability, particularly for non-benchmark VRP variants or logistics settings outside the training corpus?​​

2. Given that several domain experts found the final heuristic code verbose, redundant, or over-complicated, what mechanisms (besides code-length penalty) do you propose to systematically regularize the structure, improve succinctness, and enhance human interpretability for large LLM-generated operator populations?​​

3. What validation and error-checking processes are in place to guarantee that newly synthesized operators do not introduce infeasibility, silent bugs, or performance degradation, especially as code complexity increases over generations and as operators interact in ensembles? Is there any theoretical or empirical guarantee that VRPAgent will not produce brittle or unsafe solutions on realistic, highly constrained, or adversarial VRP instances?

---

> ### Author Response · Authors · 2025-11-21
>
> We thank the reviewer for their positive assessment of our work. We are pleased that you recognize VRPAgent as a "solid and substantial" contribution with a "reproducible pipeline," and that you value our framework's ability to outperform both handcrafted and learning-based methods using only CPU resources.
>
> We have addressed your concerns regarding code quality, component analysis, and safety below.
>
>
>
> ### Weaknesses
>
> **W1: Novelty of Heuristics (Recombination vs. New Algorithms)**
> We acknowledge that VRPAgent often discovers recombinations of existing strategies (e.g., SISRs-like removal) rather than inventing fundamentally new theoretical concepts. However, we argue that in mature fields like the VRP, "fundamentally new" components are exceedingly rare even in expert human research. State-of-the-art heuristics are almost invariably smart, novel combinations of known mechanisms.
> The core contribution of VRPAgent is automating this high-level "algorithm engineering." By intelligently organizing and adapting these components, VRPAgent creates novel operators that outperform existing SOTA baselines. We believe this capability to adapt and recombine known logic to specific problem distributions is the most practical path forward for automated heuristic design in Operations Research.
>
> **W2: Code Quality, Redundancy, and Magic Numbers**
> We agree that the raw output from the discovery phase can be verbose and difficult to read. To address this, we have introduced a **post-processing refinement step** (detailed in the revised paper).
> In this new pipeline, we separate *discovery* from *refinement*. During discovery, we encourage the LLM to generate compact, uncommented code to save tokens and costs. Once the best operator is identified, we pass it through a refinement stage where the LLM restructures the code, removes redundancy, adds comments, and clarifies "magic numbers."
> We believe this approach transforms a weakness into a major strength: unlike Neural Combinatorial Optimization (NCO) methods, which are black boxes, VRPAgent ultimately produces "white-box," human-readable code. We have provided examples of these improved heuristics in our anonymous repository: `https://anonymous.4open.science/r/vrpagent-submission`.
>
> **W3: Lack of Detailed Component Ablation**
> We agree that dissecting exactly *why* a specific ensemble works is scientifically interesting. However, we consider a granular ablation of the generated heuristic logic to be out of the scope of this paper, which focuses on the *methodology* of automated discovery rather than an OR-theoretic analysis of the resulting artifacts. A full deconstruction of the generated heuristics would require a separate study better suited for an OR journal.
> However, to address the concern about the *generalization* of findings, we added new experiments where we "warm-start" the discovery for VRPTW and PCVRP using heuristics found for CVRP. The strong convergence in these experiments confirms that the core strategies discovered by VRPAgent are robust and transferable, not just overfitting to specific benchmarks.
>
> **W4: Risk of Misconfiguration (Scattered Parameters)**
> This concern is addressed by our new post-processing step. The refinement pass organizes parameters and logic, making the code significantly easier to maintain and debug. Our expert co-authors found that the post-processed code is comparable to human-written code in terms of structure, mitigating the risk of accidental misconfiguration.
>
> **W5: Focus on Standard Benchmarks vs. Real-World Constraints**
> As one of the first works utilizing LLMs for VRP heuristic discovery, we felt it was essential to establish credibility by comparing against SOTA baselines on standard, rigorous benchmarks (CVRP, VRPTW, PCVRP). This follows standard evaluation protocols for ML/OR publications.
> We emphasize that we go beyond the simple CVRP (common in many ML papers) to include complex variants like Time Windows and Prize Collecting. Having now established that VRPAgent outperforms established baselines on these standard datasets, applying the framework to noisy or dynamic real-world constraints is a prime target for our future work.
>
> **W6: Transparency and "Black Box" Risk**
> We respectfully point out that the alternative, namely Deep Learning-based NCO, is the ultimate black box, where policies are encoded in millions of neural weights that are impossible to audit. In contrast, VRPAgent produces code. Even if the raw code is complex, it is deterministic and inspectable. With our improved post-processing, this transparency is further enhanced, allowing practitioners to verify logic before deployment, a level of safety NCO cannot offer.

---

> > ### Author Response · Authors · 2025-11-21
> >
> > **W7: Safety: Bugs, Infeasibility, and Malicious Prompting**
> > * **Feasibility:** Feasibility is guaranteed by the LNS framework's guardrails, not just the LLM code. Additionally, we employ a standard solution checker. If an operator produces an infeasible solution or crashes, it is assigned an infinite cost and naturally eliminated from the gene pool by the Genetic Algorithm.
> > * **Bugs:** The evolutionary process acts as a rigorous fuzzer; buggy code that fails to solve instances effectively is discarded.
> > * **Malicious Prompting:** We view this as a low risk for an optimization tool. VRPAgent is a backend tool for operations researchers/engineers, not a public-facing chatbot. An internal bad actor has far more direct ways to disrupt operations than by subtly prompt-engineering a heuristic generator.
> >
> > **W8: Extensibility to Packing/Scheduling**
> > **We explicitly focus on VRPs in this work**. If we inadvertently implied experimental results for packing or scheduling, it was unintentional. We have reviewed the text to ensure that we make no incorrect claims.
> >
> > ### Questions
> >
> > **Q1: Sensitivity to Initialization and GA Parameters**
> > We have added a detailed discussion and new experiments on **warm-starting** the search. We found that initializing the population for complex problems (VRPTW) using solutions found for simpler ones (CVRP) significantly accelerates convergence.
> > Regarding GA parameters, **Section 5.2** of the paper provides detailed ablations on the code-length penalty and crossover mechanisms. We found that the code-length penalty is crucial: it applies a soft pressure that keeps the logic concise without stifling the discovery of complex ensembles if they truly add value.
> >
> > **Q2: Mechanisms for Interpretability (Beyond Penalty)**
> > As detailed above, our primary mechanism for interpretability is now the **post-processing refinement step**. While the code-length penalty keeps the code compact during evolution, the post-processing step expands the code into readable, well-structured C++ code with comments and clear variable names. This ensures we get the best of both worlds: efficient search and maintainable output.
> >
> > **Q3: Validation and Error-Checking**
> > Our validation process is standard for heuristic development:
> > 1.  **Sandboxed Execution:** Operators are run within the LNS framework.
> > 2.  **Feasibility Check:** Every generated solution is passed through a rigid feasibility checker (checking capacity, time windows, etc.).
> > 3.  **Fitness Evaluation:** Infeasible solutions or runtime errors result in a fitness score of infinity.
> > 4.  **Evolutionary Selection:** The GA automatically purges brittle or unsafe operators.
> > There is no theoretical guarantee that a new heuristic will *always* find a solution (even handcrafted heuristics can fail on specific instances), but the evolutionary pressure on hundreds of training instances ensures a very high degree of robustness.

---

> > > ### Comment · Reviewer_fYZd · 2025-11-26
> > > **Helpful clarifications; core limitations persist**
> > >
> > > Thank you for the detailed and technical response. The added post‑processing/refinement pipeline, warm‑start and GA-parameter analyses, and clarification of feasibility checks and evolutionary ‘fuzzing’ substantially improve the framing and practical robustness of VRPAgent. Nonetheless, key limitations regarding deeper scientific insight into the discovered heuristics and evaluation on noisy, real‑world scenarios remain, so I thing I keep my score.

---

> > > > ### Author Response · Authors · 2025-11-26
> > > > **Plan proposal for new additions**
> > > >
> > > > Thank you for your constructive response. We are pleased that our previous revisions addressed many of your initial concerns.
> > > >
> > > > We suggest the following plan to address your two remaining concerns:
> > > >
> > > > ### 1. Quantitative Analysis of Discovered Heuristics
> > > >
> > > > We will conduct a quantitative analysis of the discovered heuristics. This involves:
> > > >
> > > > - Concept Identification: systematically identifying conceptual components (e.g., specific ordering strategies) within the heuristics
> > > > - Quantitative Assessment: measuring the impact of each component to the final performance via targeted ablations
> > > >
> > > > ### 2. Evaluation on Real-World Scenarios
> > > >
> > > > We will evaluate VRPAgent on CVRP instances with more realistic properties by extending our benchmark to include the challenging X instance set [1]. This dataset is specifically designed to cover a broad range of challenging problem structures, including instances with highly clustered customers and varying demands, thus providing a much stronger test of our agent's robustness in scenarios mimicking real-world complexity.
> > > >
> > > >
> > > > [1] Eduardo Uchoa, Diego Pecin, Artur Pessoa, Marcus Poggi, Thibaut Vidal, Anand Subramanian. "New benchmark instances for the Capacitated Vehicle Routing Problem", *European Journal of Operational Research*.
> > > >
> > > > ---
> > > >
> > > > We will begin to work on these additions immediately and aim to provide the results as soon as possible. Please let us know whether these planned experiments and analyses fully address your concerns; we remain available to make any necessary adjustments.

---

> > > > > ### Author Response · Authors · 2025-12-03
> > > > >
> > > > > While we are aware that you can not response to this message, we wanted to let you know that we have finished the promised experiments and updated the paper with the results.
> > > > >
> > > > > ### 1. Enhanced Analysis of Discovered Heuristics
> > > > > In Appendix F we now conduct an extensive ablation study for the best discovered heuristic per problem, in which we systematically remove  different subcomponents to analyze their impact on performance. This provides some interesting insights into the importantce of the different components and the inner working of the designed heuristics.
> > > > >
> > > > >
> > > > > ### 2. Evaluation on Real-World Scenarios
> > > > > We evaluated on the X instance set from CVRPLib which contains diverse instances with more real-world like properties (see Appendix C for the detailed results). The heuristic found by VRPAgent signficantly outperforms the state-of-the-art approach SISRs.

---

### Official Review · Reviewer_fmXs · 2025-10-29

**Soundness:** 3
**Presentation:** 2
**Contribution:** 3
**Rating:** 4
**Confidence:** 3

**Summary:**

Designing effective heuristics for VRP problems based on the Large Neighborhood Search (LNS) algorithm typically requires extensive human expertise and trial-and-error. To address this issue, the paper proposes using large language models (LLMs) to automatically design heuristic operators. Building on the concept of genetic algorithms, the LLM generates diverse heuristic candidates, retains the best-performing ones according to the solution results, and performs heuristic modifications and explorations to further improve performance. The proposed method is validated on multiple types of VRP problems, demonstrating a significant overall performance advantage compared with other AI-enhanced LNS approaches.

**Strengths:**

The work presented in this paper is solid and substantial, with comprehensive comparisons against many methods published in top-tier conferences, demonstrating strong overall performance.

The manuscript is well-written, logically organized, and carefully proofread.

The proposed method also shows promising potential for extension and application to other problem domains.

**Weaknesses:**

The proposed framework exhibits general applicability; however, the experimental cases are limited to the VRP domain.

The effectiveness of the proposed method still requires further investigation, as it has not been compared with widely used commercial solvers. Moreover, the results do not show a significant improvement in either computational speed or solution quality compared with existing approaches.

**Questions:**

1. In Fig. 2, why does the curve without mutation decrease faster in the early iterations, yet later perform worse than the one with mutation? It seems that 20 iterations are insufficient for convergence. It is recommended to extend the number of iterations to show a more complete convergence process.

2. The analysis of Fig. 5 (a) and (b) is inadequate. The discussion merely restates the numerical results without providing insight into the underlying reasons or the conclusions that can be drawn from them.

3. The paper claims that the proposed framework has strong transferability. Therefore, the results and implementation should be made open-source to enable further verification and application by other researchers across different problems and domains.

4. All test cases in the paper are limited to VRP-related problems, yet the method itself does not incorporate any VRP-specific structural design or analysis. Moreover, the generated heuristics are not analyzed, leaving the algorithm’s interpretability and physical rationale unclear.

5. As mentioned in Comment 4, since the algorithm is not specifically tailored to VRP, it is suggested to include additional results on other mixed-integer programming benchmarks (e.g., general MILP instances) in the appendix to demonstrate the generality and effectiveness of the proposed approach.

6. Although the paper compares with many deep learning–based methods, it does not include comparisons with classical solvers such as Gurobi or CPLEX, which are widely used in practice. Without such baselines, the practical applicability of the proposed method to real-world VRP problems remains unclear. It is recommended to supplement results comparing with Gurobi and/or CPLEX.

---

> ### Author Response · Authors · 2025-11-21
>
> We express our sincere gratitude to the reviewer for the time taken to evaluate our work. We are encouraged by your assessment that the work is "solid and substantial," "well-written," and that our method demonstrates "strong overall performance" with "promising potential" for future applications.
>
> We have carefully addressed your concerns regarding the scope of the application, the comparison to commercial solvers, and the analysis of the convergence behavior. Please find our detailed responses below.
>
> ### Weaknesses
>
> **W1: Limited Experimental Scope (VRP Domain)**
> We understand the reviewer's interest in seeing the framework applied to other domains given its generalizable appearance. However, we would like to clarify that **VRPAgent is explicitly designed and optimized for Vehicle Routing Problems**.
>
> The core of our framework relies on a Large Neighborhood Search (LNS) metaheuristic, utilizing destruction and repair operators. This structure is highly specific to permutation-based routing problems and is the primary reason our method achieves State-of-the-Art (SOTA) performance in this domain. While the *concept* of using LLMs to generate operators is generalizable, the specific architecture of VRPAgent (the "leash" we place on the agent) is tailored to VRPs. We believe that restricting the scope to VRPs allows us to demonstrate the full potential of LLM-driven discovery for a well-researched problem, rather than providing a diluted evaluation across unrelated problems.
>
> **W2: Comparison with Commercial Solvers and Magnitude of Improvement**
> We appreciate the suggestion to compare against commercial solvers to establish practical baselines. We have **added new experiments comparing VRPAgent against Gurobi**, a leading commercial MIP solver.
>
> Our results show that VRPAgent significantly outperforms Gurobi. For the CVRP, VRPAgent achieves much lower costs in the same time. It is important to note that exact solvers like Gurobi struggle to scale beyond a few hundred customers for VRP variants due to the NP-hard nature of the problem, whereas VRPAgent scales effectively to 2,000 customers.
>
> Regarding the "minor" improvement: The VRP is an exceptionally well-researched problem where decades of research have resulted in powerful heuristics like SISRs and HGS. In this context, a gap reduction of roughly **0.30%** relative to the SOTA (SISRs) is a significant achievement. In real-world logistics, where fleets cover millions of miles, a 0.3% efficiency gain translates to substantial cost and emission reductions.

---

> > ### Author Response · Authors · 2025-11-21
> >
> > ### Questions
> >
> > **Q1: Figure 2 Convergence Analysis (Mutation vs. No Mutation)**
> > The behavior observed in Figure 2 is a classic manifestation of the **exploration vs. exploitation trade-off**.
> > * **Without Mutation:** The search is purely driven by crossover and selection. This leads to a "greedy" optimization path, resulting in faster initial improvements. However, the population quickly loses diversity and converges prematurely to a local optimum.
> > * **With Mutation:** Mutation injects diversity into the elite population. While this might slow down the initial descent slightly (as not all mutations are beneficial immediately), it prevents premature convergence, allowing the algorithm to escape local optima and ultimately achieve a better final solution quality.
> >
> > Regarding the number of iterations: We limited the discovery phase to 20 iterations to keep the computational cost of the experiments reasonable for reproducibility. However, the trend is clear: the "with mutation" curve has overtaken the "no mutation" curve and continues to improve, validating the design choice.
> >
> > **Q2: Analysis of Figure 5 (Sensitivity Analysis)**
> > We thank the reviewer for pointing out the brevity of our original analysis. We have **updated the manuscript** to provide a deeper discussion of Figure 5. Specifically, we now clarify that the code length penalty creates a pressure for concise code which correlates with lower token usage (efficiency) but does not linearly correlate with performance degradation until the penalty becomes extreme. This insight helps explain why we can reduce token costs by ~50% without sacrificing solution quality.
> >
> > **Q3: Open Source and Reproducibility**
> > We fully agree that openness is key to verification. We commit to **making the full implementation of VRPAgent open-source upon acceptance**.
> > To facilitate immediate verification, we have already uploaded the discovered heuristics and all prompts to our anonymous repository: `https://anonymous.4open.science/r/vrpagent-submission`.
> >
> > **Q4: VRP Specificity and Heuristic Analysis**
> > We respectfully clarify that our method **does** incorporate VRP-specific structural design. The LNS framework (destroy/repair loop), the specific function signatures for operators (e.g., selecting customers to remove, ordering them for greedy insertion), and the evaluation environment are all deeply rooted in VRP logic.
> >
> > Regarding the analysis of generated heuristics: We have dedicated **Section 6 (Discovered Heuristic Operators)** and **Appendix H** to a detailed qualitative analysis. Three expert co-authors analyzed the code to assess readability, coherence, and novelty. They found that the LLM successfully designs complex ensemble strategies that utilize domain-specific features like spatial density and demand distribution.
> >
> > **Q5: General MILP Benchmarks**
> > As noted in our response to the first weakness, VRPAgent is not a general-purpose MILP solver; it is a specialized heuristic discovery framework for routing. Applying the current architecture to general MILP instances would not be appropriate because the underlying "safe" metaheuristic (LNS with greedy insertion) is designed for permutation problems, not general constraint satisfaction.
> > Additionally, the superior performance against Gurobi on VRP instances (discussed above) demonstrates that for this specifi class of problems, our specialized approach is far more effective than general-purpose MILP solvers.
> >
> > **Q6: Comparison with Classical Solvers (Gurobi/CPLEX)**
> > As mentioned earlier, we have incorporated **Gurobi** as a baseline in our revised experiments. The results confirm that it cannot compete with VRPAgent on the medium-to-large instances (500+ customers) that are the focus of this paper and typical of real-world applications. This highlights the practical necessity of heuristic approaches like VRPAgent.

---

### Official Review · Reviewer_7uJY · 2025-11-01

**Soundness:** 3
**Presentation:** 3
**Contribution:** 3
**Rating:** 6
**Confidence:** 4

**Summary:**

This paper presents VRPAGENT, a framework that uses Large Language Models (LLMs) to automatically discover heuristic operators for Vehicle Routing Problems (VRPs). The approach embeds LLM-generated problem-specific operators within a Large Neighborhood Search (LNS) metaheuristic and refines them through a genetic algorithm with elitism and biased crossover. The authors evaluate their method on three VRP variants (CVRP, VRPTW, PCVRP) and demonstrate state-of-the-art performance using only a single CPU core at test time.

**Strengths:**

1. Novel and Practical Framework: The approach of generating only problem-specific operators within a fixed metaheuristic is well-motivated. This "keeping AI agents on a leash" philosophy addresses key limitations of prior LLM-based approaches by ensuring correctness and manageability while still enabling discovery of novel strategies.
2. Strong Empirical Results: VRPAgent achieves impressive performance improvements over both traditional OR solvers and recent learning-based methods, with negative gaps (around -0.30%) relative to state-of-the-art SISRs on larger instances. The consistency across multiple problem variants and instance sizes is particularly compelling.
3. Computational Efficiency: The single CPU core requirement at test time is a significant practical advantage over GPU-dependent NCO methods, making deployment more accessible.
4. Thorough Experimental Analysis: The paper includes comprehensive ablation studies demonstrating the importance of biased crossover and mutation, analysis across different LLMs (showing that open-source gpt-oss achieves near-SOTA performance at low cost), and sensitivity analyses on GA hyperparameters.
5. Expert Analysis: The inclusion of expert evaluation of generated heuristics (Appendix C) provides valuable qualitative insights into readability, coherence, and novelty, adding credibility beyond pure performance metrics.

**Weaknesses:**

**Major issues**

1. Interpretability Concerns: The expert analysis consistently notes that discovered heuristics are difficult to interpret due to complex logic, nested conditionals, and convoluted use of random numbers. This limits practical adoption where transparency is important. The paper acknowledges this but doesn't propose concrete solutions.
2. Limited Generalization Analysis:
- Training is conducted only on 500-customer instances, yet the approach generalizes well to 1000 and 2000 customers. More analysis on why this generalization occurs would strengthen the paper.
- The operators are discovered separately for each problem variant. Can operators transfer across problems or be adapted more efficiently?
3. LLM Dependency:
- Best results require Gemini 2.5 Flash at ~$19 per run, which may limit accessibility
- While gpt-oss performs well, the reliance on specific LLM characteristics raises questions about reproducibility and long-term viability
4. GA Design Choices:
- The strong bias toward exploitation (80% elite in crossover, mutation only on elites) is unusual. While ablations show it works, more analysis on why exploitation is so beneficial in this search space would be valuable.
- Limited exploration of other GA hyperparameters (e.g., initial population diversity, selection mechanisms)
5. Comparison Limitations:
- Some baselines use different time budgets or hardware configurations, making direct comparison slightly less clear
- The paper compares against construction-based LLM methods (EoH, ReEvo) that don't benefit from search budgets, but limited comparison with other LLM-based improvement heuristics
6. Novelty of Discovered Heuristics: While the expert analysis confirms novelty, it also notes that heuristics are primarily "recombinations of existing ideas." The paper could better discuss what fundamentally new concepts (if any) were discovered.

**Minor Issues**
1. The abstract mentions "VRPAGENT is the first LLM-based paradigm to advance the state-of-the-art in VRPs," which is a strong claim. While the results support this, it might be beneficial to briefly acknowledge the ongoing rapid advancements in LLM-based optimization to provide full context, perhaps by rephrasing slightly to "among the first" or "a pioneering LLM-based paradigm."
2. Notation Consistency: In Algorithm 2, line 4 uses NE (non-elite) which could be confused with the elite size parameter also denoted $N_E$. Consider using different notation. In Algorithm 2, line 7 uses $RANDOM(Е)$ and line 8 uses $RANDOM(NE)$. It would be clearer to explicitly state what $E$ and $NE$ represent in this context (e.g., a list of elite individuals, a list of non-elite individuals) to avoid ambiguity for readers unfamiliar with the specific GA implementation.
3. Missing Details: The paper mentions that full prompts will be provided in the "final code release" but only shows CVRP-specific prompts in the appendix. For reproducibility, all prompts should be included.
4. Statistical Significance: Results lack error bars or significance tests, though the consistent improvements across problems suggest robustness.
5. Figure Quality: Figure 1 is informative but quite busy. Consider simplifying or providing a higher-level conceptual diagram first.

**Questions:**

1. Have you considered incorporating interpretability metrics into the fitness function to encourage more transparent heuristics without sacrificing performance?
2. Can you provide more insight into why strong exploitation (biased crossover, elite-only mutation) works so well? Is there something specific about the LLM-generated operator search space that makes this effective?
3. How sensitive is the approach to the choice of metaheuristic framework? Would similar results be achievable with other frameworks beyond LNS?
4. The expert analysis mentions ensemble approaches in all discovered heuristics. Is this a fundamental property of effective operators, or an artifact of the LLM's training or the prompt design?
5. Have you investigated whether operators discovered for one problem (e.g., CVRP) can be adapted or fine-tuned for related problems (e.g., VRPTW) more efficiently than starting from scratch?

---

> ### Author Response · Authors · 2025-11-21
>
> We thank the reviewer for their thoughtful and detailed feedback. We are pleased that you recognize the novelty and practicality of our "keeping AI agents on a leash" framework, the impressive empirical results achieving negative gaps relative to state-of-the-art (SOTA), the computational efficiency of our single-core approach, and the depth of our experimental and expert analyses.
>
> We have carefully considered your concerns and questions. Below, we address each point to clarify our contributions and detail the improvements made to the paper.
>
> ### Weaknesses
>
> **W1: Interpretability Concerns**
> We agree that interpretability is a crucial factor for practical adoption. To address this, we have introduced a **post-processing refinement step** (detailed in the revised paper) that significantly improves the readability of the discovered heuristics without compromising their performance. In this step, an LLM annotates the code, improves variable naming, and simplifies logic where possible, ensuring the final output is transparent and maintainable.
>
> We intentionally separate discovery from refinement to keep generation costs low during the evolutionary search (by generating compact, uncommented code). Furthermore, we believe the interpretability of VRPAgent is a major advantage over standard Neural Combinatorial Optimization (NCO) methods. While NCO relies on black-box deep neural networks that are nearly impossible to dissect, VRPAgent produces "white-box" algorithmic code where every decision logic can be inspected, debugged, and verified by domain experts.
>
> **W2a: Generalization from 500 to 2000 Customer Instances**
> The strong generalization of VRPAgent stems from the nature of the Large Neighborhood Search (LNS) operators. These operators function by creating and solving local neighborhoods (subproblems). The structure of these local routing decisions remains relatively stable regardless of the total problem size; effectively, the heuristic solves many small problems iteratively.
>
> The generalization capability of VRPAgent constrasts sharply with many deep learning-based NCO methods, which often learn a statistical mapping from global instance features to solutions. Such models frequently "overfit" to the training distribution (e.g., node count or density) and suffer from complexity bottlenecks (like attention mechanisms) when scaling up. Because VRPAgent discovers algorithmic logic rather than learning instance-specific distributions, it scales naturally like traditional OR heuristics. We have updated the paper to include this discussion on generalization.
>
> **W2b: Transferability of Operators Across Problems**
> We confirm that operators can be effectively adapted across problem variants. Following your suggestion, we conducted a new experiment (see **Appendix D**) where we "warm-started" the discovery process for the VRPTW and PCVRP using the best heuristic discovered for the CVRP.
>
> The results show that warm-starting significantly accelerates convergence and improves final performance, even uncovering new best heuristics for the VRPTW. This demonstrates that the LLM can successfully adapt existing algorithmic strategies to new constraints, suggesting that the cost of deploying VRPAgent on new real-world problems can be drastically reduced by reusing previously discovered operators.
>
> **W3: Cost and Accessibility (Gemini 2.5 Flash vs. Open Models)**
> While the best results were obtained with Gemini 2.5 Flash (~$19 per discovery run), we believe this cost is negligible for industrial applications where a customized SOTA heuristic can save significant operational costs. Furthermore, inference costs for proprietary models are trending downward rapidly.
>
> Crucially, we demonstrate that **open-weights models like gpt-oss** can already achieve near-SOTA performance. This ensures that our method is accessible and reproducible for the research community without relying on paid APIs. As open-weights models continue to improve (e.g., recent releases like Kimi k1.5 or DeepSeek-V3), we expect the gap between closed and open models to vanish, making high-performance heuristic discovery accessible on consumer hardware.
>
> By demonstrating success with **gpt-oss**, we ensure that our results can be reproduced years from now, regardless of API deprecations. We have provided results for both closed and open models to show the current ceiling of the method while guaranteeing that the scientific contribution remains verifiable.

---

> > ### Author Response · Authors · 2025-11-21
> >
> > **W4a: Strong Exploitation Bias in Genetic Algorithm**
> > Our heavy bias toward exploitation (80% elite retention in crossover, mutation only on elites) is empirically driven by the nature of the search space. The space of "valid C++ code" is immense, and our evaluation budget is relatively tight (approx. 1000 evaluations).
> >
> > In this context, finding a "good enough" algorithmic structure and refining it (hill climbing) proves more effective than broad exploration, which often yields invalid or poor code. This mirrors observations in traditional heuristic design, where many distinct metaheuristics (like Tabu Search vs. LNS) can all perform well if they are sufficiently optimized. We found that focusing on improving the best working code snippets yields the highest return on compute investment.
> >
> > **W4b: Hyperparameter Exploration**
> > We focused our ablation studies on the hyperparameters with the most significant impact on convergence: elite size and crossover bias. While we acknowledge that other parameters (like initial population diversity or LLM temperature) play a role, exhaustive exploration is computationally prohibitive due to the costs of LLM inference and evaluation. We believe our current analysis captures the critical drivers of our method's performance.
> >
> > **W5: Baseline Comparisons (Hardware and Time Budgets)**
> > We ran all baselines ourselves to ensure fair comparisons. The differences in hardware and time budgets reflect the fundamental nature of the methods:
> > * **NCO/Deep Learning:** Requires GPUs; running them on CPUs is often infeasible or unfairly slow.
> > * **Construction Heuristics (EoH, ReEvo):** These methods are designed to produce a single solution quickly. Extending their runtime artificially does not improve their solution quality in a way that reflects their intended use.
> > * **VRPAgent:** We use a single CPU core to demonstrate practical efficiency.
> >
> > To address the concern about comparing against improvment methods, we have added a comparison to **LLM-LNS**, a very recent LLM-driven improvement method. We implemented the CVRP extension of this method ourselves to provide a direct comparison against another search-based LLM approach.
> >
> > **W6: Novelty of Discovered Heuristics**
> > While VRPAgent did not invent a fundamentally heuristic class, we argue that this bar is exceedingly high even for human researchers. Most modern SOTA heuristics are sophisticated recombinations of existing mechanisms.
> >
> > The novelty in VRPAgent lies in the unique **ensembling and adaptations** it discovered. For instance, the expert analysis highlighted novel scoring mechanisms for customer sorting and connectivity that do not exist in the standard literature. The system automatically designs complex, problem-specific ensembles that a human would find tedious and unintuitive to construct manually.
> >
> > ### Minor Issues
> >
> > * **Abstract Claims:** We have softened the claim in the abstract to state that VRPAgent is "among the first" LLM-based paradigms to advance the state-of-the-art, acknowledging the rapid pace of the field.
> > * **Notation Consistency:** We have standardized the notation in Algorithm 2. We now use $M$ consistently for GA hyperparameters and have renamed variables to clearly distinguish between elite lists and sizes.
> > * **Missing Prompts:** We have uploaded **all** prompts, including those for VRPTW and PCVRP, to our anonymous repository (linked in the paper) to ensure full reproducibility without bloating the main text.
> > * **Statistical Significance:** In the NCO literature, results on standard large datasets (like ours) are generally considered robust without separate t-tests due to the sample size. The consistency of VRPAgent's improvement across three different problem variants and multiple sizes strongly indicates that the performance gains are statistically significant and not outliers.
> > * **Figure 1 Quality:** We acknowledge Figure 1 is dense. However, we believe it is necessary to visualize the interaction between the LLM, the GA, and the LNS. Including a second, more abstract diagram would primarily duplicate information and would exceed our available space.

---

> > > ### Author Response · Authors · 2025-11-21
> > >
> > > ### Questions
> > >
> > > **Q1: Interpretability metrics in fitness function?**
> > > We decided against including interpretability in the fitness function directly because highly readable code (with comments and verbose naming) consumes more tokens, increasing cost and latency during the search. Since performance is the primary goal, we opted for the post-processing approach described above, which achieves interpretability without penalizing the discovery process.
> > >
> > > **Q2: Why does strong exploitation work?**
> > > As noted in the "Major Issues" section, the code search space is vast. Once the system finds a valid, high-performing operator logic, it is far more efficient to refine and optimize that logic (exploitation) than to roll the dice on completely new code structures (exploration), given the limited evaluation budget.
> > >
> > > **Q3: Sensitivity to Metaheuristic Framework?**
> > > LNS is integral to our current implementation because it naturally separates "operator logic" (perfect for LLMs) from "search management" (handled by the framework). However, the core principle, using LLMs to generate specific algorithmic components within a safe harness, is transferable. We believe similar results could be achieved by learning operators for other frameworks (e.g., genetic operators for a hybrid genetic search), provided the interface is well-defined.
> > >
> > > **Q4: Are Ensembles a fundamental property or an artifact?**
> > > While our crossover prompt enables ensemble generation, they are typically longer and thus penalized by our code length penalty. The fact that ensemble-based heuristics survive and dominate despite the penalty suggests they are indeed a fundamental property of effective operators for these problems and provide robustness that single strategy operators lack.
> > >
> > > **Q5: Adaptation of operators?**
> > > Yes, as detailed in our response regarding "Transferability," we have confirmed via our new warm-start experiment that operators adapt very efficiently from CVRP to VRPTW/PCVRP.

---

### Author Response · Authors · 2025-12-03
**Summary of Revisions and Major Improvements**

Dear Area Chair and Reviewers,

We sincerely thank you for your service. We have utilized the discussion phase to make substantial improvements to the paper. We believe the revised manuscript effectively addresses the core concerns raised and significantly elevates the quality, analysis, and empirical strength of our work.

Below, we summarize the key improvements made during the rebuttal phase based on reviewer feedback:

- **SoTA Performance on CVRPLib Instances**: To demonstrate robustness beyond our synthetic datasets, we evaluated VRPAgent on the diverse X instance set from CVRPLib (Appendix C). Our discovered heuristics significantly outperform the state-of-the-art approach, SISRs, proving that VRPAgent generalizes effectively to complex, real-world distributions.

- **Enhanced Interpretability & Code Quality**: We integrated a new post-processing refinement step, as suggested by reviewers, that decouples discovery from code polishing (Section 6, Appendix F). This maintains low generation costs while producing clean, interpretable code. This highlights a critical advantage of VRPAgent over standard Neural Combinatorial Optimization (NCO): human-readable heuristics rather than opaque neural weights.

- **Analysis of Generated Heuristics**: In Appendix F, we added a new ablation study of the best-discovered heuristics. By systematically removing subcomponents, we provide concrete insights into the inner workings of the generated algorithms and the specific contribution of each component to the final performance.

- **Transferability (Warm-Starting)**: Following reviewer suggestions, we demonstrated that heuristics discovered for CVRP can successfully "warm-start" the search for VRPTW and PCVRP (Appendix E). This significantly accelerates convergence and improves final performance, validating the universality of the learned operators.

- **New Baselines**: We added Gurobi and LLM-LNS (Ye et al., 2025) as baselines to the main experiments (Table 1), ensuring our method is compared against the most relevant solvers/approaches.

- **Robustness Analysis**: We added a new success rate analysis (Appendix C) showing that the code generation maintains a consistently high validity rate (>85%) across all problem types.

You may find the changes in the revised PDF in blue color for text and with a blue rectangle for new tables and figures.

We believe that these revisions fully address the reviewers' concerns and demonstrate the value of our contribution to the community. We respectfully ask the AC to consider these major improvements in their final decision.

Best regards,

VRPAgent Authors

---

### Meta-Review · Area_Chair_5rLC · 2026-01-05

**Summary:**

This paper identifies that current LLM-based methods still fall short of producing heuristics to solve challenging combinatorial optimization problems, such as the vehicle routing problem (VRP), due to the end-to-end function design, the absence of an overall framework, and inefficient search space exploration. To address these limitations, they propose a novel LLM-based metaheuristic framework called VRPAGENT, specifically for designing heuristics for VRPs. VRPAGENT focuses on generating only the problem-specific destroy and ordering heuristic operators within a given Large Neighborhood Search (LNS) framework instead of generating an end-to-end heuristic from scratch. In addition, VRPAGENT uses a genetic algorithm (GA) with elitism and biased crossover to iteratively search for better heuristic operators. Experiments on three different VRP variants (CVRP, VRPTW, PCVRP) demonstrate VRPAGENT can discover strong heuristics that outperform both very strong handcrafted methods (SISRs, HGS, LKH3) and recent learning-based approaches.

The reviewers recognize the proposed VRPAGENT framework as "novel and practical", "solid and substantial", "clear, and the discovered operators can go beyond the expert-designed ones", and also appreciate VRPAGENT's strong performance on VRPs and the inclusion of expert analysis. However, they also have many major concerns on the current submission:

- **Interpretability:** The heuristics discovered by VRPAGENT are difficult to interpret due to complex logic, nested conditionals, and convoluted use of random numbers. (7uJY, fYZd)

- **Generalization/Transferability:** More analysis is required on whether the operators found for one specific problem can be transferred across different VRP variants. The generalization ability across different problem sizes also needs more analysis. (7uJY)

- **LLM Dependency and API Call:** Best results require Gemini 2.5 Flash at ~$19 per run, which may limit accessibility and reproducibility (7uJY). It is not clearly stated whether the comparison methods based on LLM heuristic generation have a similar number of API calls (Z5wY).

- **GA Design:** Ablations on the GA hyperparameters are not enough. The strong bias toward exploitation (80% elite in crossover, mutation only on elites) is questionable. (7uJY)

- **Experimental Comparisons:** The time budget might be unfair for different baselines. Comparisons against widely-used commercial solvers (fmXs) and more powerful LLM-based methods (7uJY,  Z5wY) on real-world problems (fYZd) are needed. More experimental analysis is needed (fYZd, Z5wY).

- **Limited Scope to VRP Only:** The experimental cases are limited to the VRP domain. (fmXs, fYZd)

- **Minor Improvement on VRP Instances:** The results do not show a significant improvement in either computational speed or solution quality compared with existing approaches. (fmXs, Z5wY)

- **Incremental Novelty of VRPAGENT:**  The evolutionary framework of VRPAgent is similar to existing heuristic evolutionary frameworks, and the proposed "code length penalty" is also negligible. (Z5wY)

- **Novelty of Discovered Heuristics:** The found heuristics are ensembles or recombinations of standard strategies from the literature, but provide little evidence of discovering fundamentally new algorithms. (7uJY, fYZd)

**Reviewer Concerns:**

The authors have made substantial improvements to the paper to address the concerns raised by the reviewers. I believe the following concerns were properly addressed by the rebuttal:

- **Interpretability:** The authors explained that the strong generalization of VRPAgent stems from the nature of the Large Neighborhood Search (LNS) operators. They have also integrated a new post-processing refinement step to decouple discovery from code polishing (Section 6) with detailed analysis and ablation on the found heuristics for each VRP variant (Appendix F).

- **Generalization/Transferability:** The authors have added a warm-start analysis to show that heuristics discovered for CVRP can successfully "warm-start" the search for VRPTW and PCVRP (Appendix E).

- **LLM Dependency and API Call:** The authors have made a valid claim that the open-weights models like gpt-oss can already achieve near-SOTA performance, which ensures the proposed method is accessible and reproducible for the research community. They also provided details about the API call to ensure a fair comparison.

- **GA Design:** The authors have provided a reasonable explanation that the heavy bias toward exploitation is empirically driven by the nature of the search space and the tight evaluation budget.

- **Experimental Comparisons:** The author has added Gurobi and LLM-LNS (Ye et al., 2025) as baselines to the main experiments (Table 1). The results show that VRPAGENT can still achieve the best performance. They also show that the heuristics discovered by VRPAgent can significantly outperform the state-of-the-art approach on real-world CVRPLib-X instances.

Nevertheless, the following concerns remain:

- **Limited Scope to VRP Only/Minor Improvement on VRP Instances/Incremental Novelty of VRPAGENT:** These three concerns are indeed closely related to each other. The authors have provided valid responses to address each of them, such as the LNS framework is highly specific for VRP, the well-researched nature of VRPs, and the "Operator-in-the-Loop" framework of VRPAGENT.

However, after reading the rebuttal, I think these concerns are still outstanding. The reviewers could still treat the proposed VRPAGENT as a minor LLM-based improvement on top of the well-studied and highly specific LNS framework for the well-researched VRP variants. I agree and appreciate the principle of "keeping AI agents on a leash" which this paper follows to design the VRPAGENT framework. However, a thorough experimental analysis of more problems beyond VRPs may be necessary to truly demonstrate the generalizability and practicality of this principle for LLM-based heuristics discovery. It is likely that the LNS framework will no longer be effective for those problems, as noted by the authors. But it is important to test the proposed idea's availability under other widely used frameworks.

- **Novelty of Discovered Heuristics:** On the contrary, it is also essential to demonstrate how the practitioner can "constrain the search to problem-specific operators nested within a robust, correctness-enforcing metaheuristic" (quoted from the principle of keeping AI agents on a leash) for real-world applications that "involve ever-changing requirements that are not supported by existing solution methods". In other words, testing the proposed idea on real-world problems where no standard strategies are available.

- **Minor Concern:** In Appendix F.2 for analysis of discovered heuristics (line 1270), it claims that "the code is generated without comments to avoid the LLM influencing the analysis". However, in line 1311 to line 1319, the Expert 2 finds the code contains "both **very meaningful high-level comments** that make it easy to quickly assess the big picture of main functions, and **low-level comments that facilitate understanding the details**." A careful, consistent check might be needed.

**Reviewer Scores:**

Taking all the above discussion into consideration, I think the reviewer 7uJY will maintain the positive score (6), reviewer fYZd could increase the score from 4 to 6 since the raised concerns have been largely addressed, while reviewer fmXs and reviewer Z5wY will keep their negative score (4 and 4) due to the remaining concerns. As a result, I recommend rejecting this paper due to the highly competitive nature of ICLR. But I highly encourage the authors to carefully address the remaining concerns and submit a revised paper to a future venue.

---

### Decision · Program_Chairs · 2026-01-26

Reject